# New insights into active tectonics and seismogenic potential of the Italian Southern Alps from vertical geodetic velocities

Letizia Anderlini[1*], Enrico Serpelloni[2], Cristiano Tolomei[2], Paolo Marco De Martini[2], Giuseppe Pezzo[2], Adriano Gualandi[3], Giorgio Spada[4]

[1] Istituto Nazionale di Geofisica e Vulcanologia, Bologna (Italy)
[2] Istituto Nazionale di Geofisica e Vulcanologia, Rome (Italy)
[3] Jet Propulsion Laboratory, California Institute of Technology (USA)
[4] DiSPeA, Urbino University (Italy)

*Correspondence to*: Letizia Anderlini (letizia.anderlini@ingv.it)

**Abstract.** This study presents and discusses horizontal and vertical geodetic velocities for a low strain-rate region of the Southalpine thrust front in northeastern Italy obtained by integrating GPS, InSAR and leveling data. The area is characterized by the presence of sub-parallel, south verging thrusts, whose seismogenic potential is still poorly known. Horizontal GPS velocities show that this sector of the Eastern Southern Alps is undergoing ~1 mm/a of NW-SE shortening associated with the Adria-Eurasia plate convergence, but the horizontal GPS velocity gradient across the mountain front provide limited constraints on the geometry and slip-rate of the several sub-parallel thrusts. In terms of vertical velocities, the three geodetic methods provide consistent results showing a positive velocity gradient, of ~1.5 mm/a, across the mountain front, which can be hardly explained solely by isostatic processes. We developed an interseismic dislocation model, whose geometry is constrained by available subsurface geological reconstructions and instrumental seismicity. While a fraction of the measured uplift can be attributed to glacial and erosional isostatic processes, our results suggest that interseismic strain accumulation at the Montello and the Bassano-Valdobbiadene thrusts are significantly contributing to the measured uplift. The seismogenic potential of the Montello thrust turns out to be smaller than that of the Bassano-Valdobbiadene fault, whose estimated parameters (LD = 9.1 km and slip-rate = 2.1 mm/a) indicate a structure capable of potentially generating a Mw > 6.5 earthquake. These results demonstrate the importance of precise vertical ground velocity data for modeling interseismic strain accumulation in slowly deforming regions, where often seismological and geomorphological evidence of active tectonics is scarce or not conclusive.

## 1 Introduction

Diffuse deformation, slow tectonic rates and long repeat times of large earthquakes make the estimate of seismic hazard at continental plate boundaries a challenging task. This is the case of the Italian Alps, a classic example of a broadly deforming continental collisional belt (Schmid et al., 2004). Present-day convergence between the Adriatic and Eurasian plates is largely accommodated in the Eastern Southern Alps, ESA (e.g. D'Agostino et al., 2005; Cheloni et al., 2014), where

the Adriatic lithosphere underthrusts the Alpine mountain belt. This area is characterized by a notable seismic risk, due to the high seismic hazard of northeastern Italy (*http://zonesismiche.mi.ingv.it*), high population density and widespread industrial activities. Geodesy indicates that the highest present-day deformation rates are localized along the mountain front of the Italian ESA (D'Agostino et al., 2005; Bechtold et al., 2009; Cheloni et al., 2014; Serpelloni et al. 2016), which is also the locus of several M>6 earthquakes (Rovida et al., 2015). For the Venetian sector of the ESA (between the Schio-Vicenza line and the Cansiglio plateau; see Fig. 1), however, historical and instrumental earthquake records indicate a lower seismic moment release rate with respect to its easternmost (Friuli) sector (Serpelloni et al., 2016). Here, discrepancies between geodetic and seismic deformation rates have been interpreted as due to partial locking of the ESA thrust front (Cheloni et al., 2014; Serpelloni et al., 2016). However, despite geological or geomorphological evidence of Quaternary deformation (Benedetti et al., 2000; Galadini et al., 2005), the identification of active faults responsible for large past earthquakes is not conclusive. This area is also affected by non-tectonic deformation transients associated with the hydrological cycle in karst regions and precipitations, which are well tracked and described by continuous GPS stations (Devoti et al., 2015; Serpelloni et al., 2018). These deformation transients, which mainly affect the horizontal components of ground displacements, make the estimate of the long-term horizontal tectonic deformation rates more challenging.

Overall, in the European Alps vertical GPS velocities correlate with topography (Serpelloni et al., 2013; Sternai et al., 2019), with widespread uplift at rates of up to ~2.0-2.5 mm/a in the North Western and Central Alps and ~1 mm/a across a continuous region from the Eastern to the South-Western Alps. Proposed mechanisms of uplift include the isostatic response to the last deglaciation (19-11 Ka BP, Clark et al., 2012), long-term erosion, detachment of the Western Alpine slab, as well as lithospheric and surface deflection due to sub-Alpine asthenospheric convection (Serpelloni et al., 2013; Chery et al., 2016; Nocquet et al., 2016). In the study area the isostatic response to last deglaciation and long-term erosion are characterized by long-wavelength sub-mm/a patterns (e.g., Sternai et al. 2012; Serpelloni et al., 2013), whereas the elastic response to present deglaciation (Barletta et al., 2006) shows localized peaks of uplift (up to the mm/a scale) far from the study area. In the Eastern Alps, active shortening contributes to the observed uplift, but the pattern and amount of tectonic uplift due to elastic strain accumulation at faults is still unknown.

In the study area ~1 mm/a of SE-NW shortening is accommodated in ~40 km (Serpelloni et al., 2016), with the largest part being accommodated across the southernmost Thiene-Bassano-Cornuda, Montello and Cansiglio thrust faults (Cheloni et al., 2014; Serpelloni et al., 2016). However, horizontal GPS velocities in the Montello and Cansiglio areas appear "noisy", providing a poor fit to a single-fault model than in the Friuli sector of the ESA front (Cheloni et al., 2014; Serpelloni et al., 2016). Barba et al. (2013) used a sparse GPS velocity field for the Alps and vertical rates from precise leveling measurements performed by the Istituto Geografico Militare (IGM; IGM-RG, 1978) in the 1952-1984 time interval, over a distance of ~100 km from the Venetian plain to the Southern Alps mountain belt. The leveling measurements show a change from negative to positive vertical rates from the Venetian plain toward the mountains, which has been interpreted by Barba et al. (2013) as associated with elastic locking of the BVT. However, the leveling line does not cross the Bassano-Valdobbiadene thrust (BVT), passing partially through the Montello thrust (MT) and then running along the Fadalto Valley

(see Fig. 2), which is interpreted by Serpelloni et al. (2016) as an active left-lateral tectonic feature, separating the Carnic from the Dolomitic Alps.

In this work we integrate InSAR and GPS ground motion rates measurements across the MT and the BVT, the southernmost portion of the ESA south-verging fold-and-thrust belt (Fig. 1), with the goal of determining a new spatially dense 3D interseismic velocity field, considering also the available IGMI leveling measurements. We use two InSAR
velocity fields obtained from ENVISAT satellite acquisitions along both ascending and descending orbits during the 2004-2010 and a GPS velocity field obtained from the analysis of data from continuous and semicontinuous stations between 1998 and 2018, with most of the GPS stations in the study area active after 2005. The datasets and the procedures used to analyze and integrate the geodetic observations are described in Section 3. In order to provide insight into the origin of the geodetic uplift we developed a two-dimensional dislocation model, jointly inverting GPS, leveling and InSAR velocities along a
NNW-SSE oriented profile crossing the MT and BVT, as described in Section 4. The results are presented in Section 5 and discussed in Section 6.

## 2. Tectonic Setting

The South Alpine belt, which is part of the Alpine, Carpathian, Dinaridic system (AlCaDi after Schmidt et al., 2008), is a SSE-vergent orogenic system formed during Neogene time by indenting one continent into another (continent-
80 continent collision). Within this model, the rigid Adriatic microplate has indented obliquely into a softer region along the southern margin of the European plate (Ratschbacher, et al., 1991a, 1991b; Frisch et al., 1998). This indenter model is consistent with the overall geometry of the Eastern Alps, with the region of maximum horizontal shortening being located north of the indenter front and with long, orogen-parallel structures that fan out toward the east and extrude the orogen eastward, toward the Pannonian Basin. The topography of the Eastern Alps also reflects the indenter tectonics causing
crustal shortening, surface uplift and erosional response (Robl et al., 2017). The present-day convergence between the Adriatic and Eurasian plates is largely accommodated in the ESA, where the Adriatic lithosphere underthrusts the Alpine mountain belt. From Serpelloni et al. (2016), the Adriatic microplate rotates counterclockwise at ~0.3 °/Myr around a pole of rotation located in the western Po Plain of Italy, implying rates of convergence between Adria and Eurasia between 1.5 and 2.0 mm/a in the study region (Fig. 1).

The Southern Alps are a typical example of a deformed passive continental margin in a mountain range (Bertotti et al., 1993). Until the Oligocene, this Adriatic domain was the gently deformed retro-wedge hinterland of the Alps, intensively reworked only at its eastern boundary by the Paleogenic Dinaric belt. From the Neogene (~23 Ma), the Southalpine fold-and-thrust belt developed and progressively propagated toward the Adriatic foreland, mainly reactivating Mesozoic (~250–65 Ma) extensional faults (Castellarin et al., 1992). The study area (dashed box in Fig. 1) is located in the Venetian Southern
Alps, which are part of the SSE-verging fold-and-thrust belt of the Southern Alps. Here the geometry of the fold-and-thrust belt is that of an imbricate fan, with a shortening of 30 km at least (Doglioni, 1992; Doglioni, 2003). The main thrusts are,

from the internal parts to the foreland, the Valsugana thrust (VS in Fig. 1), the Belluno thrust (BL in Fig. 1) and the Bassano-Valdobbiadene thrust (BVT in Fig. 1), the latter being associated with a morphological relief of ~1200 m above the plain (Fig. 3), known as Pedemountain flexure. The ESA southernmost front is now mainly buried beneath the alluvial deposits of the Venetian plain and sealed by Late Miocene to Quaternary (~ 7–2.5 Ma) deposits (Fantoni et al., 2002), and consists in the Thiene-Bassano-Cornuda thrust (TBC in Fig.1) and the MT (Benedetti et al., 2000; Fantoni et al., 2002; Galadini et al., 2005; Burrato et al., 2008; Danesi et al., 2015; MT in Fig. 1). The Montello hill (see Fig. 3) is generally interpreted as an actively growing ramp anticline on top of an active, north dipping, thrust that has migrated south of the mountain into the foreland (Benedetti et al., 2000). Benedetti et al. (2000) suggested that the Piave river changed its course because of the growth of the Montello hill, at a rate of 0.5 mm/a to 1.0 mm/a estimated from the ages of the river terraces. Modeling the deformation of the river terraces as a result of motion on the thrust ramp, Benedetti et al. (2000) estimated a constant slip rate of 1.8-2.0 mm/a.

The ESA mountain front and foreland is the locus of several M ≥ 6 earthquakes (green squares in Fig. 1) in the last 2000 years (Rovida et al., 2016), but most of the seismogenic faults associated with these large earthquakes are still debated (Database of Italian Individual Seismogenic Sources, DISS; *http://diss.rm.ingv.it/diss*). Nevertheless, there is a general consensus for some of these earthquakes that the fault sources belong to the southernmost thrust fault system, emerging at the boundary between the Venetian plane and the mountain front (Benedetti et al., 2000; Galadini et al., 2005; Burrato et al., 2008; Cheloni et al., 2014). Information from historical sources and modern earthquake catalogues show that at least 12 earthquakes with M ≥ 6 occurred in the past thousand years in the Italian Southern Alps. The last one (Mw 6.4) took place in 1976 in Friuli with several strong aftershocks (5.1 ≤ $M_l$ ≤ 6.1) occurred in the following months (Slejko et al., 1999; Pondrelli et al., 2001). The largest earthquake in the study area is the 1695 Asolo earthquake, which is estimated, from the interpretation of historical information, as a ~Mw 6.4 event. However, the seismogenic source of this important earthquake is still uncertain. Uncertainties on the location, geometry and seismic potential of active faults in the Venetian Southern Alps persist due: i) to the complex structural framework of the region, inherited by the geodynamic evolution and resulting in deformation distributed over a vast area, ii) slow deformation rates along mainly blind faults, iii) sparse instrumental seismicity, causing some areas where strong historical earthquakes occurred to appear presently almost aseismic (Fig. 1).

## 3. Geodetic Observation

### 3.1 Global Positioning System Data

We use data from all the continuous GPS (cGPS) stations available for the study region provided by public institutions and private companies, integrated by data from a denser semicontinuous (sGPS) network installed by INGV from 2009 in the Venetian Southern Alps (Danesi et al., 2015). The solution presented here is part of a wider Euro-Mediterranean geodetic solution, including ~3000 cGPS stations that are analyzed in several sub-networks, and later combined.

The GPS velocities have been estimated analyzing the position time series, realized in the IGb08 reference frame, of all stations having an observation time span longer than 5 years (in the 1998.0-2017.5 time span, in order to be consistent with IGS08 products and the reference frame), in order to minimize possible biases in the linear trend estimation due to seasonal signals (Blewitt and Lavallée, 2002) and non-seasonal hydrological deformation signals (Serpelloni et al., 2018). The daily position time series were obtained analyzing raw data with the GAMIT/GLOBK (version 10.60; Herring et al., 2015) and QOCA (*http://qoca.jpl.nasa.gov*) software, following the three-step procedure described in Serpelloni et al. (2006, 2013). Details on the GPS data processing and post-processing are provided in the Supplement, and the reader is also referred to Serpelloni et al. (2013, 2016, 2018). The horizontal and vertical GPS velocities are provided in Table S1 of the Supplement. Figure 2 shows the horizontal GPS velocities (black arrows, with 95% confidence error ellipses) in a Adria-fixed reference frame, using the rotation pole from Serpelloni et al. (2016), which has been calculated in a consistent IGb08 reference frame. The colored circles and diamonds, instead, show the vertical GPS and leveling rates, respectively, with white symbols representing stable sites (rates within ±0.5 mm/a), blue and red symbols representing points with negative (subsidence) and positive (uplift) rates, respectively.

Horizontal velocities show that the Veneto-Friuli plain is stable to within ±0.3 mm/a (see Fig. 2 and Table S1) with respect to the Adriatic microplate, whereas the ESA move toward SSE, orthogonal to the main thrust front, at rates slower than 2 mm/a. Figure 3 shows a velocity cross-section, with topography and seismicity, obtained projecting horizontal (Adria-fixed) GPS velocities along a NNW-SSE profile, highlighting the ~1 mm/a shortening accommodated between the Venetian plane and the Dolomites. Consistently with results from Serpelloni et al. (2016), the area where most of the shortening is accommodated, and the horizontal-strain rate is higher, corresponds to the southernmost thrust front (i.e. the MT, in this area). Regional kinematic models (Cheloni et al., 2014 and Serpelloni et al., 2016) show that part of the shortening across the southernmost ESA thrust front is likely accommodated by aseismic slip, although the spatial resolution of coupling maps are still poor. The horizontal velocity gradient across the Mt Grappa - Mt Cesen anticline is noisy, providing limited constraints on the presence of multiple locked faults across the mountain range.

Importantly, GPS and leveling vertical velocities coherently highlight a change from negative rates in the Venetian plain, near the Adriatic coastlines, to positive rates (up to ~1.9 mm/a) north of the Mt Grappa-Mt Cesen anticline (Fig. 2). Vertical velocities from leveling measurements refer to elevation changes measured during two campaigns in 1952 and 1984 by the Istituto Geografico Militare Italiano, IGMI (see the Supplement for additional information), suggesting that the measured uplift is a steady feature over time-scales longer than the GPS observational time span. In section 3.4 this vertical velocity gradient is better detailed and discussed.

**3.2 Synthetic Aperture Radar Data**

We use Synthetic Aperture Radar (SAR) images (C-band sensor) acquired between 2004 and 2010 by the ENVISAT satellite, operated by the European Space Agency, along both ascending and descending orbits (Fig. 2). The displacement time-series and the mean ground velocities, in the line of sight (LOS) of the satellite geometry, have been

obtained adopting the SBAS multitemporal InSAR technique (Berardino et al., 2002). The InSAR data processing has been performed using the SARScape module of ENVI software (http://sarmap.ch/tutorials/sbas_tutorial_V_2_0.pdf) to perform SBAS analysis. The SBAS algorithm is based on a combination of many SAR differential interferograms that are generated by applying constraints on the temporal and perpendicular baselines. The inversion of the interferometric phase, performed with a Singular Value Decomposition (SVD) algorithm, produces a ground displacement time series for each coherent pixel, by minimizing possible topographic, atmospheric and orbital artifacts; the achieved accuracy of the displacements can be up to ~1 mm (Ferretti et al., 2007). In case of tropospheric delays correlated with topography, we accounted for them as a phase term linearly correlated with elevation. The scaling coefficient is estimated directly from SAR interferograms, using a network approach, i.e., exploiting the redundancy of the interferometric pairs to derive a joint estimation for the scaling factor of each interferogram (Elliott et al., 2008; Wang and Wright, 2012; Jolivet et al., 2013). Further details on the InSAR data processing and post-processing are provided in the Supplement.

For descending and ascending orbits we analyzed 32 and 45 images, respectively (Table 1), selecting pairs of images applying constraints on the maximum orbital separation and the temporal distance between the two passages, in order to minimize spatial and temporal decorrelation effects. In particular, we chose 450 m and 600 days as the maximum values for the perpendicular and temporal baselines, respectively (Fig. S1 of the Supplement). We use the 90-m Shuttle Radar Topography Mission (available at *http://www2.jpl.nasa.gov/srtm*) digital elevation model for the topography subtraction step (Farr et al., 2007). The analysis of both ascending and descending datasets has been performed applying a multi-looking factor equal to 20 for the azimuth, and 4 for the range direction, resulting in a final resolution of 90 m on the ground. In order to avoid unwrapping issues and before the inversion steps, we checked each interferogram discarding all the pairs showing clear unwrapping errors and keeping the ones with low atmospheric noise. Once the unwrapped phases inversion step is completed, the displacement time series (see the time span reported in Table 1) for each coherent pixel is obtained. Then the mean ground LoS velocities have been estimated fitting, with least-squares, a linear trend to the displacement time-series of each pixel. We obtain for most of the pixels low values of Root Mean Squared Error and Chi-square (see Fig. S2 and S3 of the Supplement), indicating a mainly linear behavior for most of the dataset. The mean error associated with the linear velocity estimate is ~1 mm/a.

Table 1. ENVISAT data set. Ground resolution values are obtained after a multi-looking operation.

| Satellite | Orbit Type | Track | N. of used images | Number of pairs | Temporal span | Ground Resolution (m) | Incidence angle (°) |
|---|---|---|---|---|---|---|---|
| Envisat | Ascending | 401 | 45 | 177 | 01/03/2004 11/10/2010 | 90 | 23 |
| Envisat | Descending | 394 | 32 | 125 | 06/12/2004 06/09/2010 | 90 | 23 |

### 3.3 GPS and InSAR integration method

The GPS and InSAR velocities have been compared projecting the three components of GPS velocities along the one-dimensional SAR satellite look direction (see Fig. S4). The two datasets show a loose agreement in the velocity pattern with some discrepancies between InSAR and GPS LoS rates (mainly at the border of the frames). These differences may be due to the different reference frame (that consists on a constant offset between the two datasets) and to residual orbital errors and long-wavelength atmospheric delays that are not completely removed from the InSAR mean velocity solution (e.g., Gabriel et al., 1989; Zebker et al., 1994). These last sources of errors are commonly represented by a bidimensional planar ramp in case of short strips acquisitions, i.e., 1-2 frames (Feng et al., 2012) like in our case, or a quadratic ramp for larger InSAR images (e.g. Lohman and Simons, 2005; Biggs et al., 2007).

The difference between InSAR and GPS velocities are commonly used to correct InSAR velocities and put both geodetic datasets in the same reference frame (e.g. Zebker et al., 1994; Hammond et al., 2010). In particular, we minimize the differences between GPS and InSAR LoS rates estimating and removing a ramp signal and a constant offset for the whole InSAR dataset. We use 11 and 9 GPS sites for the ascending and descending datasets, respectively, chosen among the sites with longer time-spans (>7 years) and as much as possible uniformly distributed over each SAR frame (see Fig. 4). The InSAR velocities, for comparison with GPS LoS rates, are estimated from averaging the values from a certain number of pixels around each of the selected GPS station, testing several radiuses of selection. The optimal radius of selection is defined as the one minimizing the misfit (evaluated in terms of root-mean-square error, RMSE) between GPS and InSAR LoS rates at the selected stations after the ramp removal (see Fig. S5 in the Supplement for details). Before the correction, the RMSE deviation between GPS and InSAR data, for both the ascending and descending datasets, is of 1.15 mm/a and 0.83 mm/a, respectively. After the correction, the RMSE between GPS and InSAR datasets decreases to 0.55 mm/a and 0.63 mm/a for the ascending and descending datasets, respectively.

Figure 4 shows the corrected ascending and descending InSAR and GPS LoS velocities. The SE-NW (A-B in Fig. 4) cross sections show the overall good agreement in the LoS velocities between the two techniques. Both InSAR datasets show a positive velocity gradient of 2-3 mm/a in few tens of km across the BVT, and a common area of negative values of 1-2 mm/a in correspondence of the Venetian plain increasing toward the Adriatic coasts. Moreover, in the ascending dataset, we observe a small area with positive, but noisy, LoS velocities of ~1 mm/a across the Montello hill that are mostly within the mean uncertainty value of InSAR velocities. Figure 4 shows an area of positive LoS velocities in the northeastern corner of both the ascending and descending orbits, south of the Cansiglio plateau, which is more evident in the descending case. This area is located at the edge of both scenes and could be affected by errors associated with the orbital ramp estimation and removal during the processing chain (Berardino et al., 2002). In addition, during the filtering step (Goldstein et al., 1998, Ghulam et al., 2010) because the adopted moving window filter size was 64 pixels large, close to the frame border, where less than 64 pixels remain, the presence of artifacts is possible. However, we cannot exclude that this positive signal may be of tectonic significance, being located near the mountain front where active thrust faults are present (Galadini et al., 2005),

but the absolute values could be overestimated (as suggested by the disagreement between InSAR and GPS LoS rates near PORD) and cannot be considered reliable. Thus, we do not consider this signal in our analysis, which requires further investigations. Similarly, we excluded from the analysis the scattered pixels located in the northernmost sector of the ascending frame (Fig. 4), since they might be affected by unwrapping errors.

## 3.4 Integrated vertical velocity field

In order to estimate the vertical InSAR velocities we combine the ascending and descending LoS velocities assuming the N-S component as obtained from an interpolation of the GPS northward velocities in the area covered by the two ENVISAT frames. Given the namely near-polar sensor orbit direction, in fact, the N-S component of the InSAR velocities cannot be reliably derived. We calculate the East and Up components solving the following system of equations:

$$East = \Big((u2d/det) \times \big(Asc - (n2a * North)\big)\Big) - \Big((u2a/det) \times \big(Dsc - (n2d * North)\big)\Big)$$

$$Up = \Big((-e2a/det) \times \big(Asc - (n2a * North)\big)\Big) - \Big((e2a/det) \times \big(Dsc - (n2d * North)\big)\Big)$$

where *Asc* and *Dsc* are the ascending and descending LoS rates, *North* is the N-S interpolated component, *det = ((e2d \* u2d) - (u2a \* e2d)* and *u2d, e2d, n2d, u2a, e2a* and *n2a* are the descending and ascending direction cosines, respectively, which account for the LoS angle variability along the respective SAR images swath (Cianflone et al., 2015).

The vertical GPS and InSAR velocity field is shown in Fig. 5, together with vertical rates from the IGMI leveling measurements. Profile A-B runs normal to the strike of the MT and BVT faults, whereas the C-D profile runs along the leveling route (see the Supplement and Fig. S6 for details about how uncertainties in the leveling rates are obtained). The three geodetic observations, which refer to different temporal intervals (1952-1984 for the leveling, 2004-2010 for SAR and ~2005-2017 for GPS data, respectively), show a good agreement in the vertical ground motion rates across the two profiles. The Venetian plain, near the coast of the northern Adriatic Sea show negative rates (e.g., station MSTR), which decrease toward north, being close to zero south of the Montello hill, across which a sub-mm/a uplift is apparent from Fig. 5, but not significant, considering the uncertainties and dispersion of the geodetic measurements. North of the Montello hill, GPS, InSAR, and leveling measurements coherently show a positive vertical velocity gradient, reaching in ~20 km positive rates up to 2 mm/a. However, while this vertical velocity gradient is rather well sampled by leveling data along the Fadalto Valley (profile C-D of Fig. 5), this is not the same across the Mt Grappa-Mt Cesen anticline, where a lack of InSAR observations is present due to the highly vegetated area of the Mt Cesen-Mt Grappa chain (profile A-B of Fig. 5).

Several studies investigated subsidence processes in the Venetian plain (e.g. Carminati and Di Donato, 1999; Carminati and Martinelli, 2002; Carbognin et al., 2004; Teatini et al., 2005; Bock et al., 2012), which is due to three main causes (both of natural and anthropogenic non-tectonic origin): 1- aquifer compaction after the strong groundwater withdrawal in the second half of the last century (e.g. Gatto and Carbognin, 1981; Carbognin et al., 1995); 2- uncontrolled expansion of coastal settlements and industrial activities (e.g. Tosi et al., 2002); 3- recent sediment compaction (e.g. Brambati et al., 2003; Fontana et al., 2008). As we can see from the profile A-B of Fig. 5, subsidence rates increase from the

center of the plain towards the coasts as due to the sum of the aforementioned processes. More localized faster subsidence (> 3 mm/a) is likely due to local anthropogenic processes, mainly associated with groundwater exploitation (e.g., the area between SUSE and VITT along the profile C-D of Fig. 5). Discrepancies between GPS rates and InSAR or leveling vertical rates (e.g., stations MT02, SUSE in Fig. 5) are likely due to known problems at GPS sites (e.g., MT02 monument has been destroyed and rebuilt in a different place two times) or the onset of recent processes, over the time interval of data availability, including anthropogenic processes (e.g., SUSE is located above a gas storage area: *https://www.edisonstoccaggio.it/it/campo-collalto* and its data quality and time-series is generally poorer than other sites). Although basically no InSAR observations are available along the Mt Grappa-Mt Cesen chain and on the Cansiglio Plateau the GPS stations there (i.e. MGRD, TAMB, CANV) show vertical positive rates that are less than 0.4 mm/a. Importantly, these stations are known to be affected by non-seasonal hydrological deformation due to pressurization of fractured karst aquifers (Devoti et al., 2015; Serpelloni et al., 2018), and their motion rates may be biased by these non-tectonic, non-seasonal, transient processes.

## 4. Dislocation Modeling

In order to interpret the horizontal and vertical velocity gradients across the MT and BVT faults we develop a two dimensional fault model, constructed from geologic cross sections, seismicity and geophysical prospections (see Fig. 3). The interseismic strain accumulation is represented by slip on buried rectangular dislocations, embedded in an elastic homogeneous half-space (Okada, 1985) with a Poisson ratio of 0.25. The length of the dislocations along strike is kept long enough to avoid edge effects and the width along the dip direction is exaggerated for the deepest fault in order to mimic the far-field long-term convergence rate. This approach has been widely used to simulate interseismic deformation associated with intracontinental thrust faults (e.g., Vergne et al., 2001; Hsu et al., 2003; Grandin et al., 2012; Tsai et al., 2012; Daout et al. 2016).

We perform an inversion of the geodetic velocities on a pre-determined fault geometry (see Fig. 6), composed by: a ramp-décollement system pertaining to the Montello thrust faults (MT), which is constrained by recent instrumental seismicity (Romano et al., 2019; Fig. 3) and focal mechanisms (Anselmi et al., 2011; Danesi et al., 2015), a fault ramp representing the Bassano-Valdobbiadene thrust (BVT), which is not well depicted by instrumental seismicity but better constrained by geological and geophysical interpretations (e.g. Galadini et al., 2005; Fantoni and Franciosi, 2010; see also Fig. 3), which both connect at depth along a single deeply rooted thrust, which is extended downdip (Castellarin et al., 2006). We keep fixed the strike, perpendicular to the horizontal shortening rate direction (i.e. 65°N, see Fig. 2), the northward dip angles (see Fig. 6 for details), and the fault positions, while only for the Montello décollement and the deeper ramp the width has been kept constant. Assuming this fault geometry, we invert the geodetic velocities for determining two locking-depth (LD) values, that is the depth of the upper edge of the dislocation, indicating the shallow locked seismogenic fault portion for the MT and BVT ramps, and the dip-slip rates of all the fault segments (see Fig. 6).

The inversion method exploits a constrained, non-linear, derivative-based optimization algorithm (i.e. interior-point, see Byrd et al., 1999; Waltz et al., 2006). It allows to estimate the optimal parameter solution corresponding to a possible global minimum of the cost function, representing the misfit between the model prediction and the geodetic measurements. These algorithms depend on the gradient and higher-order derivatives in order to guide them through misfit space, thus they can get trapped in a local minimum (Cervelli et al., 2001), providing the best results when the starting point is near the global minimum. However, in order to ensure that we find a global solution in the inversion, we tested several different initial guess founding always the same model estimate. The cost function to be minimized is the weighted residual sum of squares:

$$WRSS = (d_{obs} - d_{mod})^T \times cov^{-1} \times (d_{obs} - d_{mod}) \qquad (1)$$

where *cov* is the covariance matrix of geodetic data errors, and $d_{mod}$ = **G(m)**, as **G** is the Green's function matrix depending on the fault geometry and slip parameters, *m* (Okada, 1985). The data covariance matrix *cov* is computed as follows: *cov* = $\Sigma R \Sigma^T$, where $\Sigma$ is the diagonal matrix of data uncertainty and $R$ is the data correlation matrix, that is dimensionless, equal to one along the diagonal and the off-diagonal elements representing the correlation between each couple of data. Assuming the three geodetic dataset (GPS, InSAR and leveling) independent among them, the whole covariance matrix is composed by three independent blocks, one for each dataset. The correlation values are nonzero only for the three components of each GPS site, considering the measurements obtained by the GPS stations to be uncorrelated among them, and for the leveling data, following the approach of Árnadóttir et al. (1992). The InSAR data covariance matrix is instead diagonal with equal variance of 1 mm$^2$/year$^2$ for all the pixels. The locking depths have been forced by the inversion algorithm to be within the seismogenic layer, as defined by instrumental seismicity (Fig. 3), while the dip-slip rates have been constrained to be kinematically consistent along the four fault planes. The latter constraint imposes the conservation of the horizontal and vertical motion along the fault junctions, by means of the geometric relationships proposed by Daout et al. (2016).

In the inversion we use horizontal and vertical velocities of 15 GPS sites (highlighted in Table S1 of the Supplement) located within a 35 km wide profile crossing the MT and BVT faults along a -25°N direction (dashed box of Fig. 2, perpendicular to the strike direction of the faults), selecting those stations reliable for site stability. We invert InSAR LoS rates from both ascending and descending datasets (corrected of the planar ramp), exploiting the completeness of information provided by the LoS velocities, containing both the horizontal and the vertical contributions.

InSAR velocities within the same swath profile used to select GPS stations have been subsampled in order to reduce the density of pixels while maintaining the first order ground deformation information (Fig. S7) and allowing for reasonable computational costs. Most of studies, performing a subsampling of InSAR data for geophysical inversions, use the quadtree sampling method (e.g. Jónsson et al., 2002; Pedersen et al., 2003; Lohman and Simons, 2005; Jolivet et al., 2012; Maurer and Johnson, 2014) that allows to reduce the number of data in order to represent the statistically significant portion of the displacement signal (Jónsson et al., 2002). In our case, with low deformation gradients it is highly risky to apply a subsampling method that depends on the deformation signal itself. For this reason we apply an alternative method that uniformly reduce the density of pixels and the specific technical details are provided in the Supplement. The final number of InSAR data has been reduced to less than 3000 points for each dataset (see Table S3 of the Supplement). The resampled

pixels with mean ground velocity less than -0.5 mm/a are excluded from the modeling, assuming that subsidence **is** due to non-tectonic processes that we cannot take into account in this analysis (see Section 3.4). The total number of resampled InSAR data (joining ascending and descending together) used during the inversion counts 3115 pixels. We jointly invert also the whole leveling dataset (see Table S2), calculating the associated covariance matrix as described in Árnadóttir et al. (1992), and excluding from the inversion the leveling points with vertical velocities less than -0.5 mm/a for the same reason mentioned above.

Given the larger number of InSAR velocity data, compared to the GPS and leveling ones, we apply a relative weight, *Wsar* (varying between 0 and 1), that rescales the InSAR covariance matrix. This approach allows to calibrate the relative importance of the InSAR dataset with respect to the others, aiming at finding the model that can best reproduce all the geodetic observations. The optimal value of *Wsar* (equal to 0.46) has been chosen as the "knee point" of the trade-off curves between the WRSS (Eq. 1) of GPS and InSAR data, and the WRSS of leveling and InSAR data, as the factor *Wsar* varies (see Fig. S8 in the Supplement for details). Although the weighting factor affects the InSAR covariance matrix during the inversion, the trade-off curves analysis has been performed considering the original covariance matrix of InSAR data.

## 5. Results

The estimated model parameters (locking depths and dip-slip rates), with their uncertainties, and residuals in terms of RMSE are reported in Table 2. Figure 7 shows the measured and modeled horizontal and vertical geodetic velocities, as well as the InSAR ascending and descending LoS rates. The error bounds for the estimated optimal fault parameters have been evaluated using the bootstrap method (e.g. Efron and Tibshirani, 1986; Árnadóttir and Segall, 1994) with 1000 random resampling of data. The bootstrap procedure allows us to estimate confidence intervals of the derived parameters (Segall and Davis, 1997) without making assumptions about the underlying statistics of errors (Amoruso and Crescentini, 2008), reflecting the limitations of the used data set (Cervelli et al., 2001). Figure S9 in the Supplement shows the frequency distribution of the optimal fault models after the bootstrap procedure, reporting corresponding 95 percentile confidence bounds (reported in Table 2) by means of the percentile method, and the trade-off distributions between fault parameters pairs.

The inversion results show that the Bassano Valdobbiadene thrust is characterized by a greater locking depth (LD = 9.1 km) and a faster dip-slip rate (2.1 mm/a) than the Montello ramp fault (LD = 5.6 km, SR = 0.5 mm/a). Moreover, the locking depth of the Bassano Valdobbiadene thrust is better constrained by the data than the one estimated for the Montello ramp, which is weakly constrained because of the noisy sub-mm geodetic signal (close to the geodetic techniques limits) measured above the structure, to which the fault geometry is more sensitive. The trade-off distributions between parameter pairs (Fig. S9) show that the locking depth estimates have no correlation with the other parameters, while the dip-slip rates have strict correlations among them due to the kinematic conservation constraint. The only slip-rate parameter we can consider independent is the deep ramp slip rate (SRDD, Fig. 6) representative of the far field convergence rate, and for

which we obtain the widest confidence bounds (Table 2). The obtained results point out that the surface velocity gradient is mostly explained by the BVT, thus suggesting a greater seismogenic potential than that one expected for the Montello thrust fault.

The fault model overall reproduces the observed velocity gradients from all datasets (see Fig. 7), although we find a partial discrepancy between the leveling data and the modeled vertical velocity field (Table 2, Fig. 7). This misfit may be due to the fact that the leveling line from the Montello hill runs along the Fadalto valley (see Fig. 2) and those points might be sensitive to a deformation signal associated with the Cansiglio thrust fault, located ~10 km south of the BVT (Galadini et al., 2005). In correspondence of the Belluno Valley (see Fig. 7), both GPS and InSAR vertical rates show larger misfit with respect to the fault model. In particular, InSAR data show a steeper velocity gradient than that reproduced by the model, and this may be ascribed to some artifacts not totally estimated and removed during the SAR data processing, and likely due to the correlation between atmosphere and topography gradient (e.g. Elliott et al., 2008; Doin et al., 2009; Jolivet et al., 2012; Shirzaei and Bürgmann, 2012).

Table 2. Output model parameters (with 95 percentile confidence intervals) and data residuals in terms of Root Mean Squared Errors, RMSE.

| LD Montello Ramp | LD Bassano Ramp | Slip rate Montello Ramp | Slip rate Montello Flat | Slip rate Bassano Ramp | Slip rate Deep Ramp | RMSE GPS | RMSE LEV | RMSE InSAR |
|---|---|---|---|---|---|---|---|---|
| $5.6_{-3.8}^{+3.5}$ km | $9.1_{-0.6}^{+1.3}$ km | $0.5_{-0.1}^{+0.2}$ mm/a | $0.4 \pm 0.1$ mm/a | $2.1_{-0.6}^{+0.8}$ mm/a | $2.5_{-0.7}^{+0.8}$ mm/a | 0.44 mm/a | 0.72 mm/a | 0.66 mm/a |

## 6. Discussion

Overall, our results are consistent with conclusions of Barba et al. (2013), who developed a 2D finite-element model constrained by the leveling vertical rates, showing that the amount of interseismic locking associated with the BVT must exceed that associated with the MT. The shallower locking depth estimated for the Montello ramp is also consistent with other inferences on interseismic fault coupling for the ESA front, constrained by GPS horizontal velocities, indicating the MT as partially elastically locked (Cheloni et al., 2014; Serpelloni et al., 2016).

The Bassano-Valdobbiadene fault is classified as seismogenic and capable of $M_w$ ~6.5 earthquakes in the DISS, but its late Quaternary activity is debated. No major earthquakes are unequivocally associated with this fault (see Fig. 1). The larger seismic event here, the Mw 6.4, 1695 Asolo earthquake (see Fig. 1), is located at the foothills of the mountain range delimited by the BVT front, but is in general associated with the westernmost portion of the Montello thrust, the Bassano-Corduna segment (Galadini et al., 2005; Burrato et al., 2008). Only recently local seismic networks (Priolo et al., 2015) and temporary seismic experiments (Anselmi et al., 2011; Danesi et al., 2015) have improved the imaging of faults at depth, indicating earthquakes with thrust focal mechanisms for both the Montello and Bassano-Valdobbiadene faults (Fig. 3).

During the 70s and the 80s, a large amount of data on the Pliocene-Quaternary fault activity of NE Italy have been collected. Castaldini and Panizza (1991) classified the BVT as active (i.e., showing tectonic movements in the middle Pleistocene-Holocene time). They proposed an uplift rate up to 1 mm/a, based on evidence of deformation in Würmian deposits laying on structures associated with the BVT (Pellegrini and Zanferrari, 1980; Zanferrari et al., 1980), morphotectonic evidence of scarps aligned with the thrust, offsets recorded in karsified surfaces of Mindel-Riss age, and changes in the post-Würmian hydrography (Zanferrari et al., 1980). Galadini et al. (2001) draw an updated summary of the active faults in the Eastern Southern Alps (i.e., responsible for the displacements of deposits and/or landforms related or subsequent to the last glacial maximum, LGM, set at 26-19 Ka BP at global scale following Clark et al., 2009, and more specifically at 25-21 Ka BP for the Alpine area as suggested by Monegato et al., 2017 and references therein). Galadini et al. (2001) produced a map including the main faults whose length is consistent with the occurrence of earthquakes with M≥6.2 (based on the rupture length/magnitude relationship by Wells & Coppersmith 1994), where the BVT is represented as an unsegmented active fault from the Schio-Vicenza line to the Fadalto line. However, in the most recent literature (e.g., Galadini et al., 2005; Burrato et al., 2009; Moulin and Benedetti, 2018), evidence of recent activity of the BVT appears not conclusive. It is worth considering that the lack of evidence of recent displacements in the mountainous region may have been conditioned by the erosional-depositional evolution of the Alps since the LGM. Intense erosion by the glacial tongues and subsequent rapid deposition of thick successions of alluvial deposits may have hidden the evidence of fault activity, making geomorphological evidence of recent tectonic activity debatable.

On the contrary, the recent deformation of the Montello thrust is testified by several Middle and Upper Pleistocene warped terraces of the Piave River paleo-course flanking the western termination of the fold (Benedetti et al., 2000 and references therein), and by the eastward deflection of the Piave River around the growing anticline. Benedetti et al. (2000) associated to the MT at least three earthquakes with M < 6 (778 A.D., 1268 and 1859), which have been however revised in the Italian macroseismic catalogue (Rovida et al., 2016).

Geological and geomorphological data appears thus not fully consistent with the geodetic deformation signals and the interpretation presented in this work. The slip-rate estimated for the flat and ramp portions of the Montello thrust are 0.4 ± 0.2 and 0.5 ± 0.3 mm/a, respectively. These values are smaller than those estimated from modeling the deformation of the river terraces by Benedetti et al. (2000), who estimated a constant slip-rate of 1.8-2.0 mm/a, values that are however incompatible with the sub-mm actual geodetic signal measured above the Montello hill. A possible explanation for the high rate obtained by Benedetti et al., 2000, is related to the new chronologies proposed for the Alpine LGM (as old as 25 Ka BP, Monegato et al., 2017) and for the Montebelluna megafan (125-30 Ka BP, Mozzi, 2005; Fontana et al., 2014), much older than the data available in 2000. The estimated slip-rate for the BVT (2.1 ± 0.3 mm/a) is much greater than the one reported in the DISS database (0.3-0.7 mm/a), inferred from regional tectonic and geological interpretations. To our knowledge, there are no other independent estimates of slip-rates for this fault, for which conversely our estimates are rather well constrained by a significant geodetic signal. Considering a length of ~30 km for each fault, a shear modulus of 24 GPa (as a mean value for the whole study region estimated from Vp and Vs values reported in Anselmi et al., 2011) and the estimated slip-rates

and locking depths (Table 2), the maximum magnitude expected in 1000 years as recurrence time is a Mw = 6.9 for the BVT and a Mw = 6.2 for the MT. These values should be assumed as upper bounds because, given the simple model used, we do not take into account some complexities like the presence of lateral and vertical variations of fault geometry and rheological parameters, that may prevent the co-seismic rupture propagation, and partially coupled fault portions that can reduce the seismogenic potential.

Despite the fault model provides a satisfactory explanation of the geodetic velocity gradients, it is worth considering that the vertical component of motion may be influenced by further ongoing processes on an alpine spatial scale. Indeed, GPS data show that the Alps are undergoing widespread uplift at rates up to ~2-2.5 mm/a in the North-Western and Central Alps, and ~1 mm/a across a continuous region from the Eastern to the South-Western Alps (Serpelloni et al., 2013; Chen et al., 2018), highlighting a correlation with topography (Serpelloni et al., 2013). Proposed mechanisms of uplift include isostatic response to the last deglaciation, to the present ice melting and to erosion, detachment of the Western Alpine slab, as well as lithospheric and surface deflection due to mantle convection (see Sternai et al., 2019 for a review). The relative contribution of each proposed mechanism to the geodetically measured uplift is, however, difficult to estimate, but their total contribution is of the order of few tenths of mm/a (Sternai et al., 2019). Moreover, it is also worth considering that models proposed to quantify these contributions (e.g. Barletta et al., 2006; Spada et al. 2009; Norton and Hampel, 2010; Sternai et al, 2012; Mey et al., 2016) are less reliable at the borders of the Alpine belt (Sternai et al., 2019), where our study area is located, making it even more speculative to apply a plausible "isostatic" correction to the actual uplift rates. The spatial patterns associated with the aforementioned processes have a characteristic wavelength of hundreds of km, and correcting for these contribution would slightly reduce the intensity of uplift rates, leaving the observed velocity gradient mostly unchanged. In order to evaluate the effects of "correcting" the actual vertical velocities for long wavelength isostatic contributions on slip-rate and locking-depth estimates, we remove a linear gradient of uplift rate along 100 km of distance from the measured geodetic rates. Despite being speculative and based on strong assumptions (see the Supplement for details), we can assume this approach as an upper-bound case to evaluate the impact of correcting or not for long-wavelength uplift signals on the inversion results. After the application of this correction, the locking-depth and slip-rate estimates are slightly reduced (see Figure S10 in the Supplement) without significant variations, resulting widely within the error bounds of the optimal fault parameters (Table 2) and leading to exactly the same seismogenic potential estimates for the MT and BVT faults.

As these long-wavelength processes could hardly explain the local vertical velocity gradient observed across the Venetian mountain front, a possible source of more localized uplift rate gradient in the study area may come from the contribution of local ice or from present-day glacier shrinkage. At the LGM the Belluno valley hosted the Piave glacier up to an elevation of ~1150 m (Pellegrini et al., 2005; Carton et al., 2009). Pellegrini and Zambrano (1979) have estimated a width of ~15 km for the ice pond, a length of ~45 km and a maximum thickness of 600-800 m. Given the spatial correlation between the fastest vertical geodetic velocities and the location of the former Piave ice pond, we have employed a simple isostatic model with the aim of providing a first-order estimate of the spatial pattern and range of magnitudes for the isostatic

response to deglaciation of this local ice sheet. The isostatic response has been evaluated using the TABOO software (Spada et al., 2004, 2011), assuming a cap profile for the pond, an equivalent area of 45 x 15 km$^2$, a maximum thickness of 800 m (Pellegrini et al., 2005), and an instantaneous deglaciation occurring 10 ka ago. This is an upper bound case since recent

studies (Pellegrini et al., 2005; Carton et al., 2009) evaluate the complete disappearance of the glacier occurred likely before 15 ka BP. The Earth model includes an elastic crust and a mantle viscosity profile consistent with the ICE-5G(VM2) model of Peltier et al. (2004). Figure S11 in the Supplement shows how the expected vertical velocities due to the Piave ice melting are sensitive to the assumed crustal thickness. Assuming in this region a crustal thickness between 30 and 40 km (Molinari et al., 2015), the model predicts modest glacial isostatic uplift rates (~0.1 mm/a).

Although we acknowledge the importance of glacio-isostatic contributions to the present-day uplift of Eastern Southern Alps, the vertical velocity gradient identified by terrestrial and space geodetic measurements across the venetian mountain front is mostly associated with active tectonic processes. Indeed, a simple elastic dislocation model based on seismological, geological and geophysical evidence can satisfactorily explain both the horizontal GPS velocities and the vertical GPS+InSAR+leveling rates, consistently. However, important constraints from new seismicity data and new surface

and subsurface geological observations will be required, together with denser GNSS data, to better constrain the tectonic rates and seismogenic potential in this area.

## 7. Conclusions

      We study the interseismic deformation at the Adria-Alps boundary in northeastern Italy, focusing on a sector of the Southern Alps fold-and-thrust belt, where WSW-ENE oriented sub-parallel, south-verging, thrusts presently accommodate

~1 mm/a of NW-SE shortening. Seismogenic faults databases indicate two possible sources for the larger historical earthquakes of the area: the Bassano-Valdobbiadene thrust (BVT), bordering the mountain front, and the Montello thrust (MT), buried beneath the alluvial venetian plain. Both GPS and InSAR measurements, consistently with precise leveling measurements, show a positive vertical velocity gradient from the Venetian plain toward the Alps, with the highest rates, of the order ot 1.5 mm/a, located ~20 km north of the mountain front. The three geodetic measurements span different time-

periods from 1950 to the present day, suggesting that this uplift pattern is consistent over a longer time interval than the most recent GPS measurements.

      We develop an interseismic fault model, constrained by InSAR, GPS and leveling rates, estimating slip-rates and locking depths of the BVT and MT faults, whose geometry is defined by geological and seismological data. This simple model satisfactorily explains both horizontal and vertical geodetic velocities, suggesting that both thrust faults are

accumulating elastic strain. The higher locking-depth and slip-rate estimated for the BVT, which is required to explain the observed uplift rate gradient, point toward a higher seismogenic potential of this fault (compatible with a $Mw_{max}$ > 6.5 in 1 Kyrs), accordingly to recent results indicating a limited interseismic locking for the MT frontal ramp.

This work demonstrates the important constraints that vertical ground velocity measurements can provide for interseismic deformation studies associated with thrust faults in slowly deforming regions, as already shown in other faster

deforming convergent boundaries. More importantly, since several processes contribute to mountains uplift, with glacio-isostatic contribution being the greatest ones in the Alps, a more accurate estimate of the contribution of each process at local scale is necessary in order to correct the measured geodetic uplift for its non-tectonic components, guaranteeing a more reliable estimate of the geodetic slip-rates. From this point of view, the Eastern Southern Alps represent an ideal natural laboratory for further investigations.


*Data availability*. GPS and leveling velocity data that support the findings of this study can be found in the Supplement. The InSAR line-of-sight velocities obtained in this work are available through the B2SHARE service within the European EUDAT infrastructure at: https://doi.org/10.23728/b2share.486d3b553c564cc6826e24548e85ad1d

*Supplement*. The supplement related to this article is available on-line at:

*Author contributions*. ES processed the GPS data, CT and GP processed the InSAR data and PMDM provides the leveling data. AG wrote the code for the InSAR data subsampling and LA developed the data analysis, integration and correction of all the datasets, and carried out the data modeling. GS developed the GIA analysis. LA, ES, CT and GS interpreted the

results and wrote the article. All the authors have contributed in the article preparation.

*Competing interests*. The authors declare that they have no conflict of interest.

*Acknowledgements*. We are grateful to all public and private institutions and companies that make GPS data freely available

for scientific applications. In particular, we acknowledge the EPN-EUREF network, Leica-ITALPOS, Topcon-NETGEO, ASI-GEODAF, INGV-RING, InOGS-FREDNET, Rete "Antonio Marussi" in Friuli Venezia Giulia, STPOS and TPOS in Trentino-Alto Adige and Regione Veneto, and the ARPAV-Belluno. Most of the figures have been created using the Generic Mapping Tools (GMT) software (Wessel & Smith, 1998). Mario Anselmi is thanked for fruitful discussions about the modeling and the analysis of results. We thank Romain Jolivet, Giovanni Monegato and an anonymous reviewer for their

constructive comments that allowed to improve this manuscript.

*Financial support*. LA is supported by INGV-MISE-DMGS 2018-19 project. GS is funded by a FFABR (Finanziamento delle Attività Base di Ricerca) grant of MIUR (Ministero dell'Istruzione, dell'Università e della Ricerca) and by a DiSPeA research grant. This study has been partially supported by the project TRANSIENTI, founded by the MIUR "Premiale

2014", by the Italian Presidenza del Consiglio dei Ministri, Dipartimento di Protezione Civile (Proj. S1 2013-2014 INGV-

DPC Agreement), and by the MIUR "Pianeta Dinamico" institutional project. This paper does not necessarily represent an official opinion or policy of the Dipartimento di Protezione Civile.

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

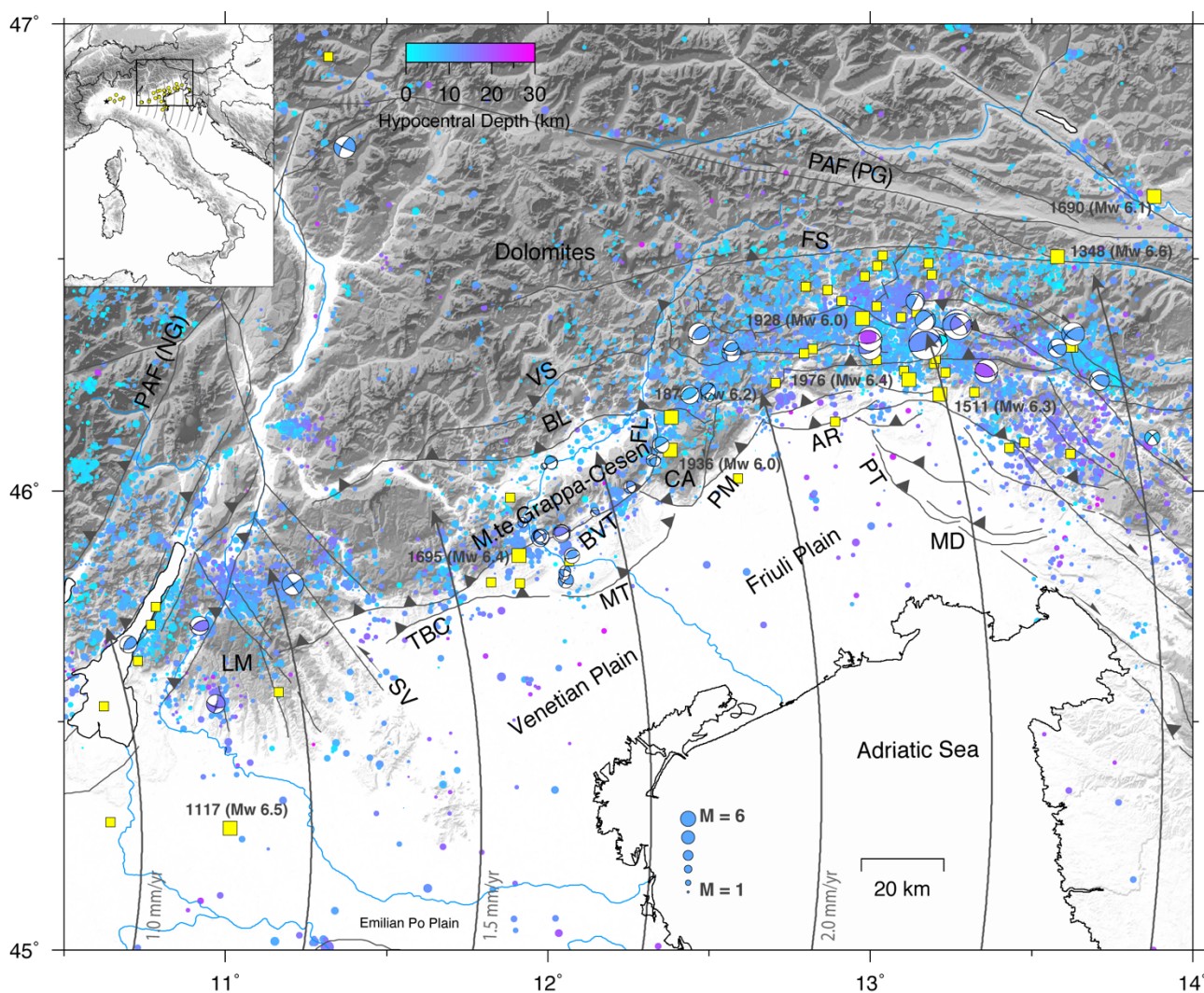

**Figure 1: Seismotectonic map of the study area, with the major tectonic lineaments shown in black (from Serpelloni et al., 2016). LM: Lessini Mountains; SV: Schio-Vicenza line; TBC: Thiene-Bassano-Cornuda thrust; MT: Montello thrust; BVT: Bassano-Valdobbiadene thrust; BL: Belluno thrust; VS: Valsugana thrust; CA: Cansiglio plateau; PM: Polcenigo-Maniago thrust; AR: Araba-Ragona thrust; PT: Pozzuolo thrust; MD: Medea thrust, FS: Fella-Sava line; PAF (PG): Periadriatic fault - Pusteri-Gailtal line. Major historical events (Mw>5 small squares and Mw>6 greater squares) from the CPTI catalog (Rovida et al., 2016) are shown as yellow squares. The blue-purple symbols show instrumental seismicity for the 2000-2017 time span extracted from the OGS bulletins (http://www.crs.inogs.it/bollettino/RSFVG). The focal mechanisms (from Anselmi et al. 2011; Danesi et al., 2015; Serpelloni et al., 2016) are plotted with the same color palette of the OGS instrumental seismicity. The small circles around the geodetic pole of rotation showing the motion direction of Adria relative to Eurasia (Serpelloni et al., 2016). Insert map: Location of the Euler rotation pole (star), GNSS stations (yellow dots) used to define the Adria rotation pole in Serpelloni et al. (2016) and small circles of the Adria-Eurasia rotation pole with respect to the region of interest (black box).**

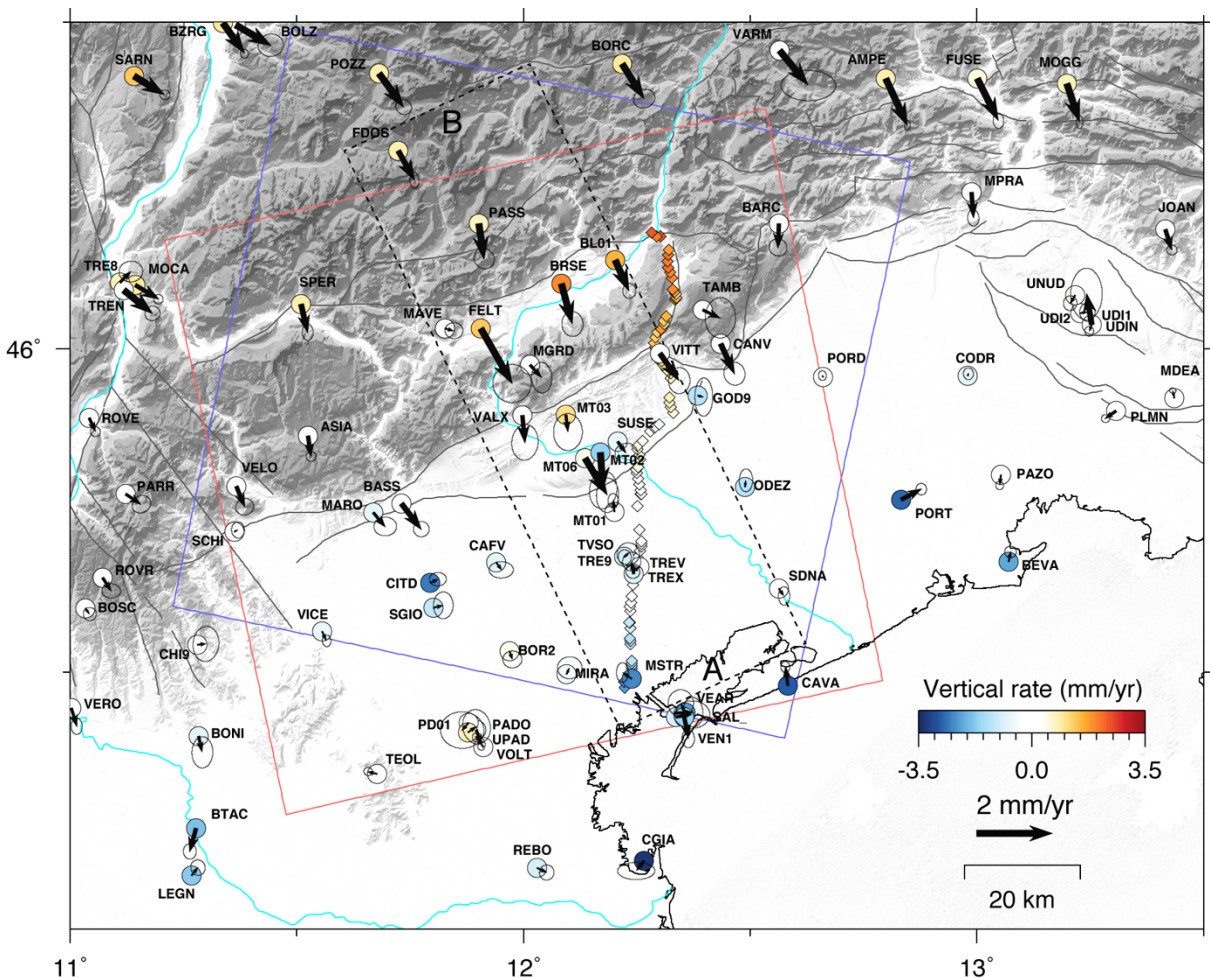

**Figure 2: Geodetic dataset: the black arrows show the horizontal GPS velocities (with 95% error ellipses) in a fixed-Adria reference frame, whereas the colored circles show the vertical GPS velocities, where blue colors indicate negative (subsidence) values and red colors indicate positive (uplift) values. The colored diamonds show the vertical ground motion rates estimated from the IGM leveling data in the 1952-1984 time interval, using the same color scales of the GPS velocities. The blue and red squares indicate the ascending and descending ENVISAT frames, respectively, used in this work. The dashed box shows the A-B cross section shown in Fig. 3.**

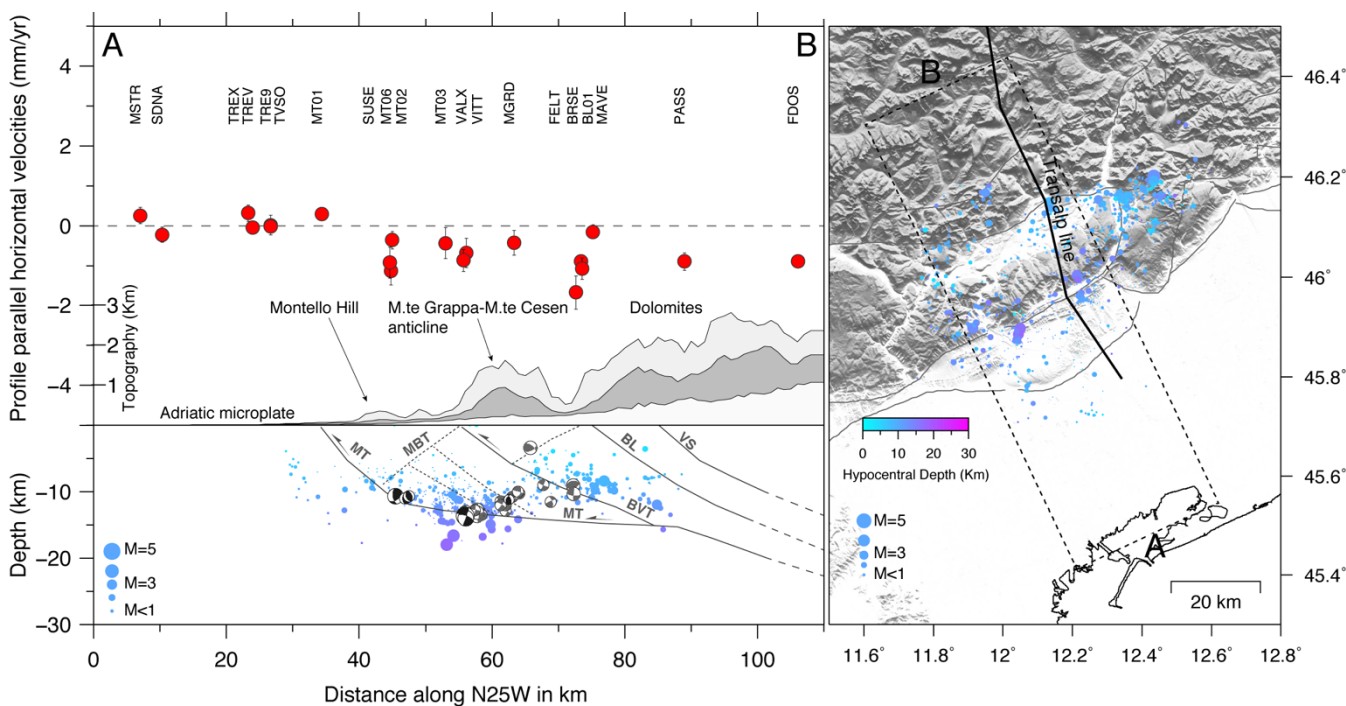

Figure 3: [left panel] Cross section of GPS velocities, topography, instrumental seismicity and focal mechanisms included in the dashed box of Fig. 2 (and the right panel). Profile-parallel velocity components, in a fixed-Adria reference frame (see Fig. 2), are shown as red circles, with 1σ error bars in gray. The dark gray areas show the average (median) topography in the profile swath, with the light gray and white areas showing the maximum and minimum elevations, respectively. The blue-purple circles show instrumental seismicity, in the 2012-2017 time interval, relocated from Romano et al. (2019), as function of depth and scaled with magnitude. The gray and black focal mechanisms are from Anselmi et al. (2011) and Danesi et al. (2015), respectively. The gray continuous and dashed lines represent major and secondary faults digitized from the TRANSALP profile interpretation (modified from Fig. 11 of Castellarin et al., 2006). MT = Montello thrust, MBT = Montello backthrust, BV = Bassano–Valdobbiadene thrust, BL = Belluno thrust, VS = Valsugana thrust. [right panel] Map showing the instrumental seismicity recorded by the Collalto seismic network (Priolo et al., 2015) relocated by Romano et al. (2019), the swath profile (dashed box) and the TRANSALP line (black line).

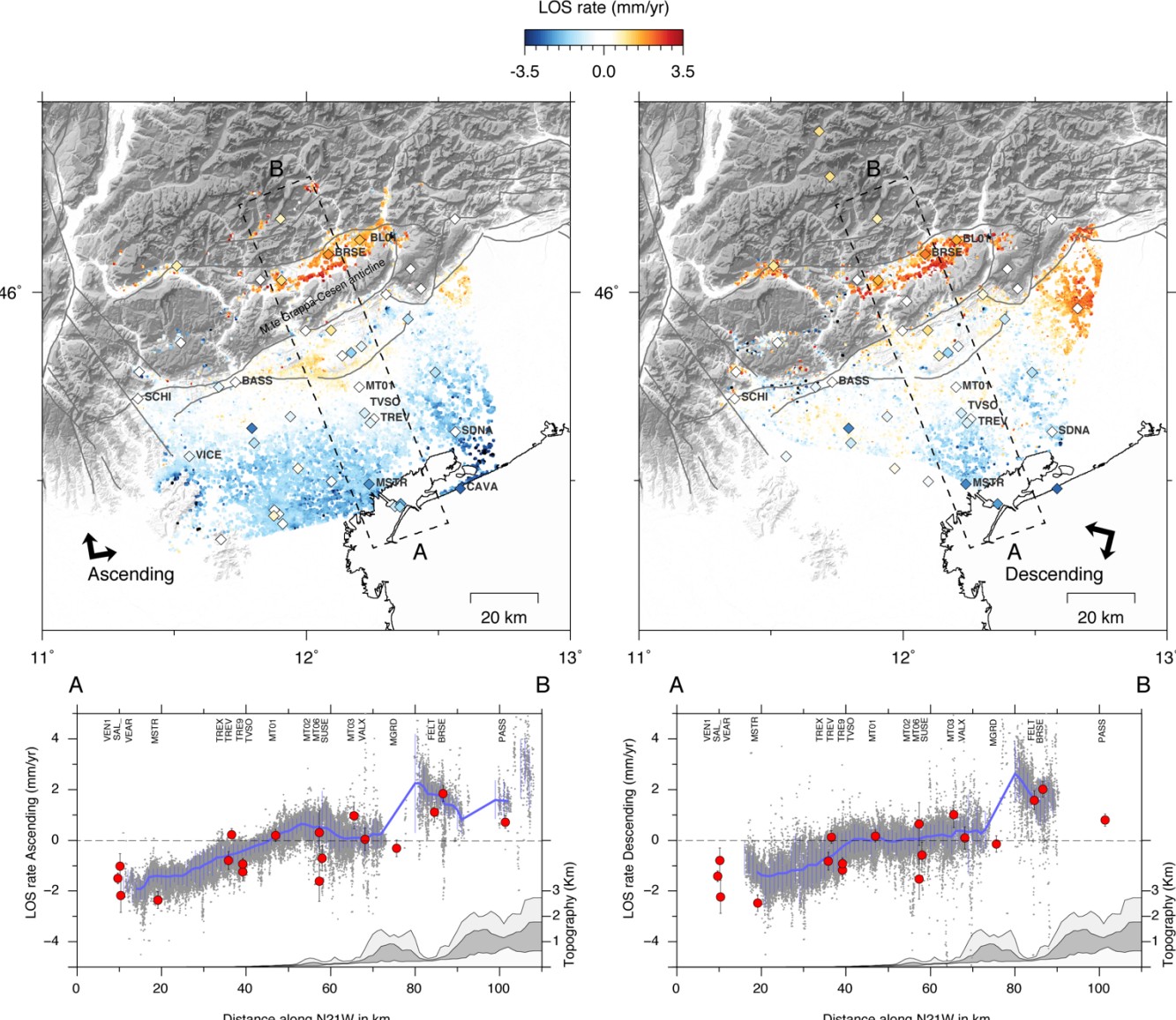

**Figure 4: InSAR line-of-sight (LoS) velocities (after the ramp removal, see Section 3.3 of the main text for details) for the ascending (left) and descending (right) orbits, with negative (blue) and positive (red) values indicating increasing and decreasing distance between the Earth surface and the satellite. Colored diamonds indicate the 3D GPS velocities projected along the SAR LoS directions. Bottom panels show the SAR and GPS LoS velocities (red circles with 1σ error bars in gray) along the A-B cross section (dashed box) for the ascending and descending orbits, respectively. The blue line indicates the median value of the InSAR data along cross section, given the minimum number of 50 SAR pixels in each bin of 1 km, and the light bars represent the data dispersion for each bin.**

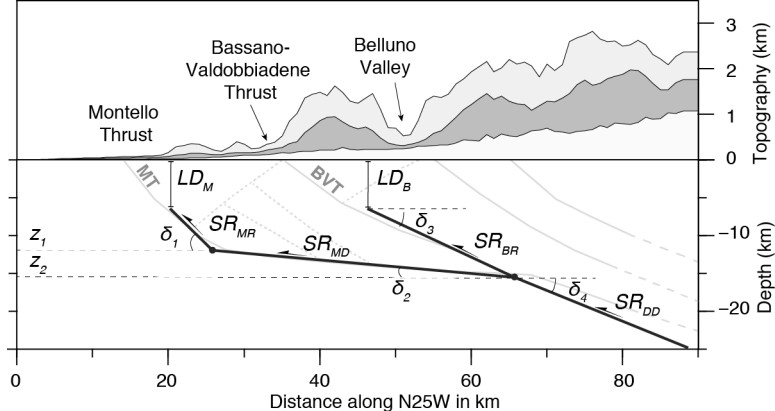

**Figure 5: Left: Vertical velocity field from InSAR data decomposition (dots), from GPS measurements (circles) and from the IGMI leveling data (diamonds), where blue colors indicate negative (subsidence) values and red colors indicate positive (uplift) values. Right: cross sections of vertical velocities along the A-B (top) and the C-D profile (bottom). The blue lines and red circles are the same as in Fig. 4.**

**Figure 6: Sketch of the geometries for two-dimensional fault model as described in Section 4. LDM (locking depth of Montello fault), LDB (locking depth of Bassano fault), SRMR (slip rate of Montello Ramp), SRMD (slip rate of Montello Decollement), SRBR (slip rate of Bassano Ramp) and SRDD (slip rate of the Deep Ramp) are the fault parameters estimated from the data inversion (see Table 2). The dip angles of each fault segment ($\delta 1 = 45°$, $\delta 2 = 5°$, $\delta 3 = 25°$, $\delta 4 = 22°$) and the depth of fault junctions ($z1 = 12$ km, $z2 = 15.5$ km) have been keep fixed.**

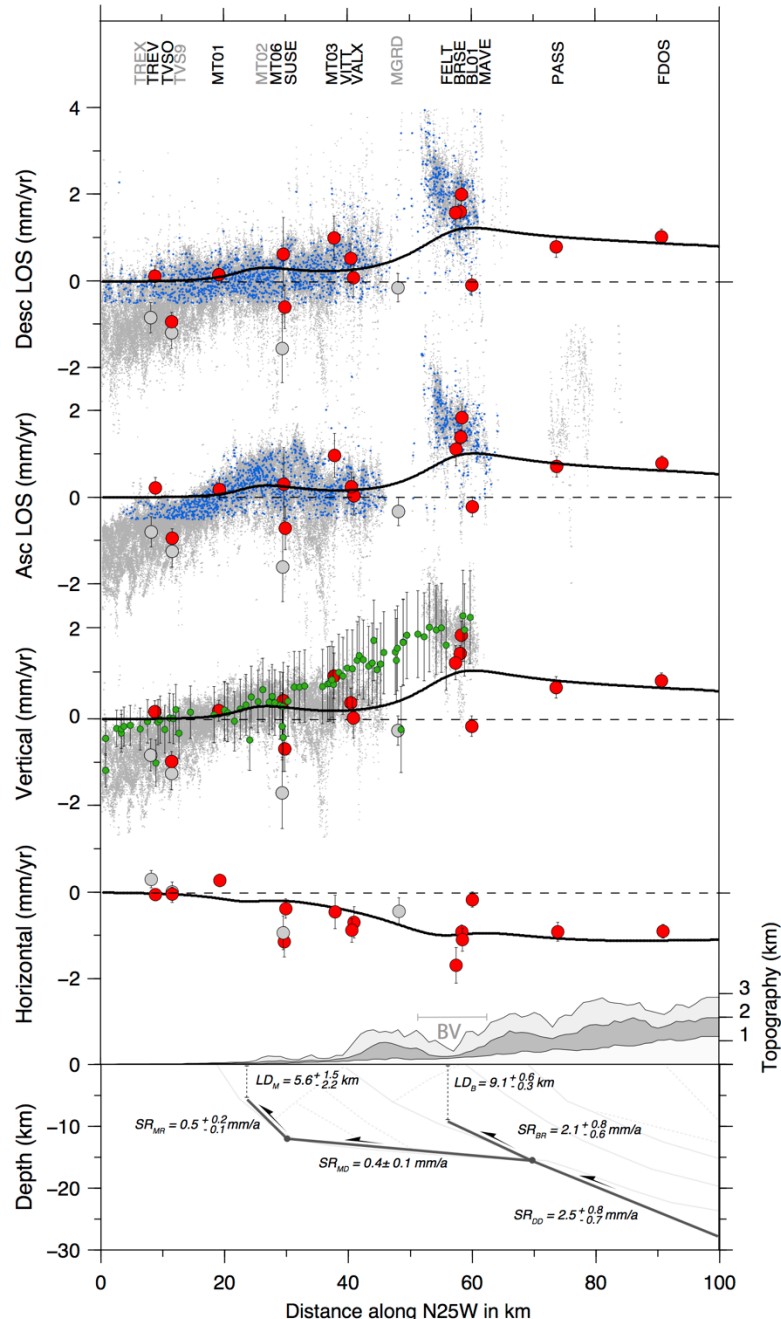

**Figure 7: Cross sections, across the A-B profile in Fig. 2, showing the modeled (black lines) horizontal and vertical velocities, as well as the SAR ascending and descending LoS rates, along with the measured ones. Gray-colored GPS sites have not been used in the inversion, unlike the red ones. G–green points indicate leveling data and small blue dots represent the subsampled InSAR LOS rates used during the inversion, displayed along with the un-subsampled InSAR dataset (light gray points). The bottom panel**

**reports the optimal fault geometry with dip-slip rates and locking depths estimates and 95 percentile confidence bounds (see Section 5 for details). BV: Belluno valley.**