# Peer review of "New insights into active tectonics and seismogenic potential of the Italian Southern Alps from vertical geodetic velocities"

_Solid Earth, 2020_

## Referee Comment (RC1) · Anonymous Referee #1 · 8 Apr 2020

**Review of SE-manuscript Anderlini et al. 2020**

The study regards a topic of importance and interest and the manuscript in general is well structured and well written. The study area along the southern boundary of the Italian Eastern Southern Alps (ESA) is known for a high seismic hazard in combination with a low-to-moderate plate tectonic strain that is difficult to locally pinpoint to specific faults. The paper reviews the various publications that recently documented different parts of the geodetically determined (mainly horizontal) strain and the seismic deformation rates that exhibit significant discrepancies across the ESA thrust fault system. As the title of the manuscript implies, new insight is gained by combining the existing tectonic, seismic and geodetic information with additional vertical geodetic velocity data integrated from GPS, InSAR and leveling measurements.

I congratulate the authors to a good study and manuscript. I do have only a few general points for consideration by the authors and a number of smaller issues with some of the figures that I list below. Overall, I believe the entity of these issues would require just MODERATE REVISION.

In the first of their main chapters –chapter 3 Geodetic Observations- the authors provide not only a careful evaluation and combination of InSAR and GPS observations but also a discussion of the different geodetic measurement techniques and their resulting data. This certainly denotes a very useful commented summary review of geodetic techniques also for the non-specialist readership. In the subsequent second main chapter of the paper, the authors develop a 2D fault model to represent the ESA front fault system and to interpret the horizontal and vertical strain gradients measured across the fault system. Again the modelling is characterized by careful evaluation and balanced weighting of the different geodetic data. While the model seems to generally „reproduce the observed velocity gradients from all (geodetic) data sets", „geological and geomorphological data appears not to be fully consistent" and I would also list the seismologic information as not being fully consistent. I fully agree with the conclusions by the authors, but I do believe that in correspondance with the careful evaluation and interpretations of the fit and the discrepancies (1) between different geodetic data sets and (2) between the best-fit 2D fault model and the geodetic data, an additional conclusion would not only be appropriate but necessary: we need better seismicity and better geological subsurface structure information. The former obviously does not understandably correspond with the geodetic data assembly and the latter is very poorly constrained or entirely speculative (see also comment below).

The seismicity shown in Figures 1 and 3 suffers from several questions regarding the consistency. (1) the time periods are not the same but do overlap between 2012 and 2017. Are you showing events from both data sets for this overlap time? (2) How to the events of the two data sets compare during this overlap time? (3) How comes much fewer events yellow-red are seen within broken line box in Fig. 1 than visible in cross section in Fig. 3? (4) Where does the obvious cluster seen in Fig. 3 at 10km depth between 45km and 65km profile distance locate in Fig1 map view? (5) Why do you project the seismicity across a band

50km wide onto a profile where you project the geodetic data only across a band 20km wide? How do the fault plane solutions compare with each other, with the geologic fault geometries (Fig. 3) and with your final model (Fig. 7)?

In Figure 3 you show the subsurface geometries of the ESA frontal fault system and as reference you refer in Figure caption and in text (lines 255 and 267) to Castellarin et al. 2006 („The gray continuous and dashed lines represent major and secondary faults digitized from the TRANSALP profile interpretation (modified from Castellarin et al., 2006). MT = Montello thrust, MBT = Montello backthrust, BV = Bassano–Valdobbiadene thrust, BL = Belluno thrust, VS = Valsugana thrust. "). In Castellarin et al. 2006 Fig. 7 is showing a geologic interpretation of vibroseismic image (down to 5km) and in Fig.8 a geologic interpretation of seismic time section again down to 5km and these figures document a profile that runs across the Montello Hill within the study region of this paper. These high-resolution seismic images and their geologic interpretations significantly differ in even the most prominent geometries with the interpretation shown in Fig. 3 of this study. Rather, in Castellarin et al. 2006 Fig. 11 (see included figure below) entitled „simplified general interpretation oft he TRANSALP profile there is shown a major detachment fault system separating and translating the imbricated upper crust at about 10km-12km depth from the middle and lower continental crust of Adria. It is the geometry of these fault

[Figure]

Fig. 11 Castellarin et al. 2006

system shown in red solid lines that seems to have been used for the model shown in Figure 3 of the current study. It seems difficult though to correlate the reflectivity image in fig.11 with the presented fault interpretation since (1) the reflectivity ends at 8km depth, (2) the shallow parts of the fault cut across well-documented continuous seismic signals and (3) no seismic evidence is visible for the detachment fault system at 10km to 12km depth. It is difficult to understand why the local high-resolution images and geologic interpretation would be ignored and a „simplified general interpretation" of regional scale would be used for a local study like the current one. You must provide details of what and why you „modified from Castellarin et al. 2006" and you must refer to the precise Figure that you used from Castellarin et al. 2006 and provide reasons for your specific choice. Finally, it seems the major change you imposed regards the

introduction of the SRDD (Figs. 6 and 7) that seems to play a major role in your model. However, please consider that the introduction of this and the stipulated other parallel faults dipping down to 20km are purly speculative as no evidence in the TRANSALP seismic data can be found in Castellarin et al. 2006 and in addition such fault contradict the concept and model documented in Fig. 11 with a pronounced and important subhorizontal detachment fault at 10-12km depth reaching far to the North beneath ESA.

**Specific comments:**

Figure 1. Much to busy figure. Reduce opacity of topography grey. Sizes of circles blue-purple and yellow-red reflect magnitude, please show scale. Rotation of Adria relative to Europe – what exactly is used as stable Europe relative point? What portion of Adria is rotating – your inset suggests all of Adria but this is difficult to justify for westernmost Adria. Explain why two different periods 2000-2017 and 2012-2018 are combined and what does this mean for the seismicity to be representative for?

Figure 2. increase size of colored circles. If Adria is rotating relative to Europe as shown in Fig. 1, what would be the local motions of the stations within ESA relative to the rotating Adria look like?

Figure 3. red dots and their uncertainty estimates: since profile runs oblique to rotation minor circles of Adria, do these uncertainty estimates include the relative differences of rotating Adria?
Note that the seismicity shown along the profile AB extends beyond the dashed box shown in Fig.2, box in Fig. 2 should be as long as AB profile in Fig. 1 and 3. What are the hypocenter location uncertainties? Are the hypocenter parameters of the two earthquake data sets calculated with the same velocity model, with the same magnitude? Please add hypocenter depth color codes as in Fig. 1. Regarding geometry of proposed fault system see critical comment above.

Figure4. „(after the ramp removal)" please explain or refer to text. bottom panels please refer to red dots in figure caption.

Figure 6. Regarding geometry of subsurfae model see critical comment above.

---

## Referee Comment (RC2) · Romain Jolivet (Referee) · 10 Apr 2020

In this article, Anderlini et al propose to apply an approach that has been applied extensively to various tectonically active regions globally but, to my knowledge, not very often to the actively deforming areas in the alps. The authors first derive some velocity fields from GNSS and InSAR data and describe some available leveling measurements. They propose a decomposition of the InSAR velocity maps into vertical and horizontal velocity fields, which are then discussed. They move on to a very classic 2D elastic modeling of the deformation to explore potential stress accumulation when considering the active faults in the region.

[Figure]

In general, the paper is well written and I do not see major issues with it. However, some points need to be discussed and my comments might require a bit of work. Figures are clear (although texts could be emphasized on the maps). I see three main issues in the paper that require being fixed before publication but, after that is done, this paper will be a very interesting contribution to the discussion on how active are these frontal thrusts surrounding the Alps. I hence recommend moderate revisions and I am looking forward to see a revised version of the article. I have set major revisions in the review system because there is no intermediate step between minor and major for this journal.

Main Comments:

- There is very little discussion on how the selection of the data is performed to avoid the effect of subsidence in the plain. The authors propose a strict threshold of -0.5 mm/yr of vertical motion below which any deformation is considered as subsidence and removed from the data fed into the model. In my opinion, this is risky, as some long wavelength subsidence might affect the general pattern of deformation. If subsidence is high near the coast and in the plain, as implied by the data, then there should be a bending effect that will affect the whole dataset. The wavelength of such bending might depend on the processes at stake, but it is unlikely that a strict threshold will allow to bypass this discussion. My point mainly arises from the fact that (and this is an issue) your model does not really fit the InSAR and leveling data you are using. The relatively high rates of uplift measured in the north are not correctly predicted by your model (which underdetermines uplift) while the low rates to the south are over-determined. It seems that there is a constant trend between the geodetic data and the model. Geodetic data agree well with each other, which is great, but the model does not really manage to catch up. This could also be caused by isostatic adjustment adding a long wavelength deformation (i.e. a wavelength longer than profile you have established). One possibility would be to explore the effect of a linear trend (or whatever long-wavelength pattern you can think of) that would represent the long wavelength

deformation needed on top of what results from dislocations in an elastic half space. This requires exploring the tradeoff between this long wavelength deformation signal and what is predicted in terms of locking depth and slip rates for both faults. It should have an impact and should be accounted for in the inverse problem.

- There is not enough details on how the InSAR data have been processed. Although the SBAS method is now quite known, quantitative information is required to assess the quality of the velocity field. It is not only because it correlates quite well with GPS that everything has been done right. For instance, correcting for tropospheric delays using a phase-topography correlation when trying to unravel a signal that correlates as well with topography is dangerous. One could easily mix deformation with tropospheric delays. Furthermore, since the region has quite strong topographic gradients, unwrapping is probably challenging and there is not a word on that (which method is used for unwrapping? In general, which software is used to compute the interferograms?). Would it be possible to see a baseline plot? Also, is there connectivity issues within the network, considering potential unwrapping issues? What is the RMS of the reconstruction of your time series? How linear is the time series? Is there a time dependent signal? There is much more details provided for the processing of GPS data and the processing of InSAR being much less standardized than GPS these days (especially with the old Envisat data) suggests there is a lot to be added in the manuscript. Finally, a lot of people have developed comparable methods for InSAR downsampling and they deserve some credit (see Lohman & Simons 2005, Jolivet et al 2012, 2015 or Sudhaus & Jonsonn 2009 for instance, but there is many other papers mentioning this).

- The description of the inversion procedure is incomplete. The algorithm used to find the minimum of the cost function should be, at least, named. Furthermore, I suspect there is some regularization of the inverse problem involved (maybe not), but please mention it. In addition, the data covariance is not described. How is it determined? One cannot follow the deal with weights if one cannot reconstruct the covariance matrix. Then, there is a problem in the a posteriori covariance discussion. The authors mention the a posteriori covariance is derived for the linear terms while bootstrap is used for the non-linear terms. In my opinion, the covariance that is derived here is obtained considering the least squares criterion (without regularization? with regularization? Is it just GˆT Cdˆ{-1} G ?) but then, it only corresponds to a "slice" of the model space, that slice corresponding to the best non-linear parameters obtained. If so, the a posteriori covariance is greatly underestimated as it is only representative of a joint marginal of the full a posteriori PDF. Finally, one can see in supplementary material figure S6 that the range of possible models for the locking on the Montello Ramp is bi-modal. Then, if it is not Gaussian, why choosing the mean model? It seems that some models could be more appropriate. Would it be possible to sample for all the possible models using a Monte Carlo approach, which would give all the tradeoffs between the various parameters (and potentially solve the issue raised in my first comment)?

For the minor comments, please refer to the annotated pdf I have sent along with my review. Looking forward to read an improved manuscript, if I am required to do so. I also strongly encourage the authors to add their geodetic data (i.e. the GPS, InSAR and leveling rates presented in the paper) to an online repository so other scientists can have a go at the modeling, once this study is published.

Romain Jolivet, PhD

Please also note the supplement to this comment:
https://www.solid-earth-discuss.net/se-2020-10/se-2020-10-RC2-supplement.pdf

**Supplement:**

[revised manuscript text omitted]

---

## Short Comment (SC1) · 14 Apr 2020

Dear Authors of the manuscript "New insights into active tectonics and seismogenic potential of the Italian Southern Alps from vertical geodetic velocities". This article is interesting in providing new geodetic data about deformations in this important sector of the Southern Alps. My remarks are related to the general approach that, here and in many similar works, the chronology and Quaternary stratigraphy are used for deformation rates assessment without using updated information about the landscape and sedimentary evolution of the study area. This happens because no geomorphologists or Quaternary stratigraphers are normally involved in such kind of studies. However,

because of the importance of the age assessment is a key for the conclusions of these works, a more careful attention has to be paid. In the present work, many chronological references sound obsolete for the innovative scope of the research.

I listed the specific concerns for this work as follow:

Line 48: the "last deglaciation" is meant after the Last Glacial Maximum I would include the time span (19-11 ka BP according to Clark et al., 2012), for the Alps see also Ivy-Ochs (2015).

Line 52: Spada et al. (2009) is referred to the "last deglaciation" but not to the "present deglaciation" that refers to the ice waning after the Little Ice Age.

Lines 345-350: the use of old and updated terminology (Mindel-Riss-Würm) from old literature shows how the authors have not updated their knowledge about the Pleistocene stratigraphy of the study area. This was update in a review article by Carton et al. (2009), who provided a long list of radiocarbon data for the Piave catchment.

Lines 353-354: the LGM is not at 20-18 ka but at 26-21 ka. I suggest to use Clark et al. (2009) for a global LGM or Monegato et al. (2017) for the Alpine LGM or Rossato et al. (2018) that gave the most updated chronology for the Veneto Prealps and piedmont plain. Also Pellegrini et al. (2005) included a useful chronology for the Belluno and Cesen-Col Visentin sector.

Line 366: maybe the Geological and geomorphological data appears thus not fully consistent just because the Authors need to use the proper updated dataset.

Lines 368-369: Benedetti et al. (2000) rates for the Montello are affected by the wrong age estimation of the Montebelluna megafan, whose age was already given as pre-LGM by Fontana et al. (2008, 2014). I suggest to use these references for this discussion.

Lines 394-395: Benedetti et al. (2000) cannot be used for assessing the thickness of the Piave glacier in Belluno, because the work is not about Pleistocene glaciations.

Pellegrini et al. (2005) and Carton et al. (2009) have to be considered instead. The Piave glacier was not an ice-sheet, which has to be used for the Alps (i.e., Alpine Ice Sheet).

Line 401: this is age is unreliable, the deglaciation in the Piave catchment occur after 18-17 ka, see Pellegrini et al. (2005) and Carton et al. (2009). At 10 ka valley glaciers were already waned. The collapse of the Alpine glaciers was fast but it occurred at 18-17 ka as reported also by Ravazzi et al. (2014) and Rossato and Mozzi (2016).

References

Benedetti, L., Tapponnier, P., King, G. C., Meyer, B., & Manighetti, I. (2000). Growth folding and active thrusting in the Montello region, Veneto, northern Italy. Journal of Geophysical Research, 105(B1), 739–766.

Carton, A., Bondesan, A., Fontana, A., Meneghel, M., Miola, A., Mozzi, P., Primon, S. & Surian, N. (2009). Geomorphological evolution and sediment transfer in the Piave River watershed (north-eastern Italy) since the LGM. Géomorphologié: Relief. Process. Environ. 3, 37–58.

Clark, P. U., Dyke, A. S., Shakun, J. D., Carlson, A. E., Clark, J., Wohlfarth, B., Mitrovica, J. X., Hostetler, S. W. & McCabe, A. M. (2009). The Last Glacial Maximum. Science 325, 710–714.

Clark, P.U., Shakun, J.D., Baker, P.A., Bartlein, P.J., Brewer, S., Brook, E., Carlson, A.E., Cheng, H., Kaufman, D.S., Liu, Z., Marchitto, T.M., Mix, A.C., Morrill, C., Otto-Bliesner, B.L., Pahnke, K., Russell, J.M., Whitlock, C., Adkins, J.F., Blois, J.L., Clark, J., Colman, S.M., Curry, W.B., Flower, B.P., He, F., Johnson, T.C., Lynch-Stieglitz, J., Markgraf, V., McManus, J., Mitrovica, J.X., Moreno, P.I. & Williams, J.W. (2012). Global climate evolution during the last deglaciation. Proc. Natl. Acad. Sci. 109 (19), E1134–E1142.

Fontana, A., Mozzi, P., & Bondesan, A. (2008). Alluvial megafans in the Veneto-
Friuli Plain: Evidence of aggrading and erosive phases during Late Pleistocene and Holocene. Quaternary International, 189(1), 71–90.

Fontana, A., Mozzi, P., & Marchetti, M. (2014b). Alluvial fans and megafans along the southern side of the Alps. Sedimentary Geology, 301, 150–171.

Ivy-Ochs, S. (2015). Glacier variations in the European Alps at the end of the last glaciation. Cuadernos de Investigación Geográfica 41, 295– 315.

Monegato, G., Scardia, G., Hajdas, I., Rizzini, F. & Piccin, A. (2017). The Alpine LGM in the boreal ice-sheets game. Sci. Rep. 7, 2078.

Pellegrini, G.B., Albanese, D., Bertoldi, R. & Surian, N. (2005). La deglaciazione alpina nel Vallone Bellunese, Alpi meridionali orientali. Geografia Fisica e Dinamica Quaternaria. Supplemento 7, 271–280.

Ravazzi, C., Pini, R., Badino, F., De Amicis, M., Londeix, L. & Reimer, P. (2014). The latest LGM culmination of the Garda Glacier (Italian Alps) and the onset of glacial termination. Age of glacial collapse and vegetation chronosequence. Quat. Sci. Rev. 105, 26–47.

Rossato, S. & Mozzi, P. (2016). Inferring LGM sedimentary and climatic changes in the southern Eastern Alps foreland through the analysis of a 14C ages database (Brenta megafan, Italy). Quaternary Science Reviews, 148, 115-127.

Rossato, S., Carraro, A., Monegato, G., Mozzi, P., & Tateo, F. (2018). Glacial dynamics in pre-Alpine narrow valleys during the Last Glacial Maximum inferred by lowland fluvial records (northeast Italy). Earth Surface Dynamics, 6, 809-828.

Best Wishes Giovanni Monegato

---

## Author Comment (AC1) · 21 May 2020

**Response to "Review of SE-manuscript Anderlini et al. 2020"**

**We are grateful for the constructive comments provided by the anonymous referee. He/she raises two main concerns, i.e. the seismic catalogs used and the fault geometry definition, which are addressed in detail below. Furthermore, specific comments to individual points of the manuscript are provided. In the following, we will repeat the referee's statements and our reply to it (in bold font).**

The study regards a topic of importance and interest and the manuscript in general is well structured and well written. The study area along the southern boundary of the Italian Eastern Southern Alps (ESA) is known for a high seismic hazard in combination with a low-to-moderate plate tectonic strain that is difficult to locally pinpoint to specific faults. The paper reviews the various publications that recently documented different parts of the geodetically determined (mainly horizontal) strain and the seismic deformation rates that exhibit significant discrepancies across the ESA thrust fault system. As the title of the manuscript implies, new insight is gained by combining the existing tectonic, seismic and geodetic information with additional vertical geodetic velocity data integrated from GPS, InSAR and leveling measurements.

I congratulate the authors to a good study and manuscript. I do have only a few general points for consideration by the authors and a number of smaller issues with some of the figures that I list below. Overall, I believe the entity of these issues would require just MODERATE REVISION.

In the first of their main chapters –chapter 3 Geodetic Observations- the authors provide not only a careful evaluation and combination of InSAR and GPS observations but also a discussion of the different geodetic measurement techniques and their resulting data. This certainly denotes a very useful commented summary review of geodetic techniques also for the non-specialist readership. In the subsequent second main chapter of the paper, the authors develop a 2D fault model to represent the ESA front fault system and to interpret the horizontal and vertical strain gradients measured across the fault system. Again the modelling is characterized by careful evaluation and balanced weighting of the different geodetic data. While the model seems to generally „reproduce the observed velocity gradients from all (geodetic) data sets", „geological and geomorphological data appears not to be fully consistent" and I would also list the seismologic information as not being fully consistent. I fully agree with the conclusions by the authors, but I do believe that in correspondence with the careful evaluation and interpretations of the fit and the discrepancies (1) between different geodetic data sets and (2) between the best-fit 2D fault model and the geodetic data, an additional conclusion would not only be appropriate but necessary: we need better seismicity and better geological subsurface structure information. The former obviously does not understandably correspond with the geodetic data assembly and the latter is very poorly constrained or entirely speculative (see also comment below).

**We agree with the reviewer, additional new geological (surface and subsurface) and seismological data are certainly required to better interpret the geodetic signal and we add this sentence:**

**"However, important constraints from new seismicity data and new surface and subsurface geological observations will be required, together with denser GNSS data, to better constrain the tectonic rates and seismogenic potential in this area."**

The seismicity shown in Figures 1 and 3 suffers from several questions regarding the consistency. (1) the time periods are not the same but do overlap between 2012 and 2017. Are you showing events from both data sets for this overlap time? (2) How to the events of the two data sets compare during this overlap time?

**We thank the reviewer for this comment that gives us the opportunity to fix an issue that can generate confusion in the readers. We agree that the use of two seismic catalogues, with spatial and temporal overlaps is misleading. We intended to use the two catalogues for illustration purposes, but we acknowledge that this is causing confusion.**
**Yes, there is overlap between the two catalogues, and the events present in both catalogues (for the overlapping period, 2012-2017) are plotted twice, in slightly different locations (and different colors). The intent of the figure was to acknowledge the presence of this high-resolution seismic catalogue, presented in a recent work (Romano et al., 2019) that has been used to provide a "Portrait of the Montello Thrust". We do not aim at any quantitative comparisons between seismic and geodetic strain rates. We have modified figures 1 and 3, plotting only the OGS bulletin data in Fig. 1, to give a regional overview of the spatial distribution of instrumental seismicity from 2000, and the high-resolution catalogue from Romano et al. (2019), in the 2012-2017 time-interval, in Fig. 3, for comparison with subsurface structures, focal mechanisms and model geometry.**

(3) How comes much fewer events yellow-red are seen within broken line box in Fig. 1 than visible in cross section in Fig. 3? (4) Where does the obvious cluster seen in Fig. 3 at 10km depth between 45km and 65km profile distance locate in Fig1 map view?

**The seismicity overlaps in Fig. 1 and in Fig. 3, but the symbols (scaled with magnitude) in Fig. 1 are much smaller than in Fig. 3. The two figures are now re-drawn considering this comment and the previous one.**

(5) Why do you project the seismicity across a band 50km wide onto a profile where you project the geodetic data only across a band 20km wide?

**The GPS velocities are projected along a 35 km wide profile (see Section 4), not 20, and the same is done for seismicity, for which we plotted the earthquakes in the same swath profile. We have re-drawn this figure considering only Romano et al. (2019) earthquakes. The dashed boxes in Fig. 2 and Fig. 3 are the same, and used to project GPS velocities and plot instrumental seismicity. We removed the dashed box in Fig. 1 that could generate confusion.**

How do the fault plane solutions compare with each other, with the geologic fault geometries (Fig. 3) and with your final model (Fig. 7)?

**In general we see some correlation between thrust faulting mechanisms and geological structures. The focal mechanisms plotted in this figure come from two local and temporary experiments (Anselmi et al. 2011 and Danesi et al. 2015). We acknowledge, and better state in the Discussion section of the manuscript, that the hypocentral depths suffer for the use of different seismic velocity models. In particular, low-dipping thrust events on the Montello flat (from Danesi et al., 2015) are considered reliable, and used to constraint the depth of the decollement in the dislocation model.**

In Figure 3 you show the subsurface geometries of the ESA frontal fault system and as reference you refer in Figure caption and in text (lines 255 and 267) to Castellarin et al. 2006 („The gray continuous and dashed lines represent major and secondary faults digitized from the TRANSALP profile interpretation (modified from Castellarin et al., 2006). MT = Montello thrust, MBT = Montello backthrust, BV = Bassano–Valdobbiadene thrust, BL = Belluno thrust, VS = Valsugana thrust. "). In Castellarin et al. 2006 Fig. 7 is showing a geologic interpretation of vibroseismic image (down to 5km) and in Fig.8 a geologic interpretation of seismic time section again down to 5km and these figures document a profile that runs across the Montello Hill within the study region of this paper. These high-resolution seismic images and their geologic interpretations significantly differ in even the most prominent geometries with the interpretation shown in Fig. 3 of this study.

**In figure 7 and 8 of Castellarin et al., 2006 the vertical scale is in seconds (5 s) probably TWT but it is not specified, therefore we suppose the sections presented are not depth migrated. In this case it is difficult to retrieve from these figures the exact geometry of the drawn fault lines, since the rocks involved are quite different from metamorphic to carbonatic and sandstone units characterized by a large variability in seismic waves velocities (ca 3-8 km/s). Differently, Figure 11 of Castellarin et al., 2006 presents a simplified general interpretation of the TRANSALP profile (TRANSALP Working Group, 2002), it is depth migrated, and thus it provides a real interpreted geometry for the tectonic structures presented even if with less details.**

Rather, in Castellarin et al. 2006 Fig. 11 (see included figure below) entitled „simplified general interpretation of the TRANSALP profile there is shown a major detachment fault system separating and translating the imbricated upper crust at about 10km-12km depth from the middle and lower continental crust of Adria. It is the geometry of these fault

[Figure]

Fig. 11 Castellarin et al. 2006

system shown in red solid lines that seems to have been used for the model shown in Figure 3 of the current study. It seems difficult though to correlate the reflectivity image in fig.11 with the presented fault interpretation since (1) the reflectivity ends at 8km depth, (2) the shallow parts of the fault cut across well-documented continuous seismic signals and (3) no seismic evidence is visible for the detachment fault system at 10km to 12km depth. It is difficult to understand why the local high-resolution images and geologic interpretation would be ignored and a „simplified general interpretation" of regional scale would be used for a local study like the current one. You must provide details of what and why you „modified from Castellarin et al. 2006" and you must refer to the precise Figure that you used from Castellarin et al. 2006 and provide reasons for your specific choice.

**As the reviewer rightly intuited, we used Figure 11 of Castellarin et al. 2006 for the fault geometries proposed in Figure 3. We are aware of the limits of its interpretation, but the profile is appropriate for the purposes of the modeling approach. Indeed in the context of the interseismic deformation modeling, the shallow portion of the faults, that are supposed to be locked, are not responsible of the observed deformation that is conversely due to the deep aseismically creeping fault portions (e.g. Vergne et al., 2001; Daout et al., 2006). For this reason our model is not influenced by the specific geometries of shallow fault structures, but on the contrary by the deep detachment fault system that aseismically slips. Moreover the used geometric interpretation is in agreement with the recorded seismicity and the available focal mechanisms (Figure 3) providing important constraints on the fault decollement. For completeness we provide below the comparison of the geometric sketch shown on Figure 6 of the manuscript with the part of Figure 11 of Castellarin et al. (2006), that we used to set up the fault geometry. We are aware that seismic interpretation have wide margins of error but the constraints provided by the instrumental seismicity reinforce the chosen model.**

[Figure]

Here we compare the same sketch (Figure 6) with the local high-resolution geological interpretation provided by Galadini et al. (2005) in Figure 11, proving that the proposed geometry is also compatible with this surficial geological cross section.

[Figure]

Finally, it seems the major change you imposed regards the introduction of the SRDD (Figs. 6 and 7) that seems to play a major role in your model. However, please consider that the introduction of this and the stipulated other parallel faults dipping down to 20km are purly speculative as no evidence in the TRANSALP seismic data can be found in Castellarin et al. 2006 and in addition such fault contradict the concept and model documented in Fig. 11 with a pronounced and important subhorizontal detachment fault at 10-12km depth reaching far to the North beneath ESA.

**As we show in the sketch above, the proposed fault geometry is in overall agreement with the interpretation documented in Fig. 11 of Castellarin et al. (2006), where the considered deep detachment faults follow the geometry indicated by the profile. The Deep Decollement (DD) is derived by the interpretation of TRANSALP profile that highlights at ~15 km depth a possible deepening of the basement rocks, that is however at the border of our model, where we have few data, playing a marginal role in the fault system. Anyway we agree that faults dipping down to 20km from the TRANSALP interpretation are speculative and that these structures might be affected by wide uncertainties.**

Specific comments:

Figure 1. Much to busy figure. Reduce opacity of topography grey. Sizes of circles blue-purple and yellow-red reflect magnitude, please show scale. Rotation of Adria relative to Europe – what exactly is used as stable Europe relative point? What portion of Adria is rotating – your inset suggests all of Adria but this is difficult to justify for westernmost Adria. Explain why two different periods 2000-2017 and 2012-2018 are combined and what does this mean for the seismicity to be representative for?

**As stated above, we fix the issue removing from this figure the seismicity from Romano et al. (2019), yellow-red symbols, that covers the 2012-2017 period. We followed all the graphical suggestions. As regard the Adria-Eurasia relative rotation, we add in the inset the position of the GPS stations used to determine the Adria rotation pole in Serpelloni**

**et al., (2016), where as well are reported all the site used to represent the stable Eurasian plate.**

Figure 2. increase size of colored circles. If Adria is rotating relative to Europe as shown in Fig. 1, what would be the local motions of the stations within ESA relative to the rotating Adria look like?

**Ok, we modified the figure accordingly to the comment. The figure already shows the horizontal velocities of GPS stations with respect to the Adria plate. Indeed the GPS stations located in the Venetian, Friuli and Emilian Po Plains (see Figure 1 for geographical references) show almost zero horizontal velocities being representative of a stable plate (in this specific reference frame, of course).**

Figure 3. red dots and their uncertainty estimates: since profile runs oblique to rotation minor circles of Adria, do these uncertainty estimates include the relative differences of rotating Adria?

**Yes, the uncertainties are projected along the profile direction.**

Note that the seismicity shown along the profile AB extends beyond the dashed box shown in Fig.2, box in Fig. 2 should be as long as AB profile in Fig. 1 and 3. What are the hypocenter location uncertainties?

**The box in Fig. 1 was shown only for indicating the study area, but we have now removed from the figure to avoid confusion. In Figure 3 we now plot only the high-resolution earthquake catalogue from Romano et al. (2019), in the 2012-2017 time-interval, selecting all the events for which hypocentral uncertainties are provided, which are of the order of 1-2 km. In particular more than 90% of the events have the horizontal and vertical errors (ERH and ERZ) less than 1.5 km and 3 km, respectively, and the whole dataset has a mean value of ERH = 0.7 km and of ERZ = 1.1 km.**

Are the hypocenter parameters of the two earthquake data sets calculated with the same velocity model, with the same magnitude?

**No, the velocity models are different, and since the one used by Romano et al. (2019) is more appropriate for the study region, we decide now to show in this figure only these earthquakes, and not those ones from the OGS bulletin.**

Please add hypocenter depth color codes as in Fig. 1. Regarding geometry of proposed fault system see critical comment above.

**Ok, done.**

Figure4. „(after the ramp removal)" please explain or refer to text. bottom panels please refer to red dots in figure caption.

**Ok, done**

Figure 6. Regarding geometry of subsurface model see critical comment above.

**For the proposed geometry see the answer reported for the comment above.**

**References**

**Anselmi, M., Govoni, A., De Gori, P., and Chiarabba, C.: Seismicity and velocity structures along the south-Alpine thrust front of the Venetian Alps (NE-Italy), Tectonophysics 513, 37–48, doi:10.1016/j.tecto.2011.09.023, 2011.**

**Castellarin, A., Vai, G.B., and Cantelli, L.: The Alpine evolution of the Southern Alps around the Giudicarie faults: A Late Cretaceous to Early Eocene transfer zone, Tectonophysics, 414, 203–223, doi:10.1016/j.tecto.2005.10.019, 2006.**

**Danesi, S., Pondrelli, S., Salimbeni, S., Cavaliere, A., Serpelloni, E., Danecek, P., Lovati, S., and Massa, M.: Active deformation and seismicity in the Southern Alps (Italy): The Montello hill as a case study, Tectonophysics 653, 95–108, doi:10.1016/j.tecto.2015.03.028, 2015.**

**Daout, S., Barbot, S., Peltzer, G., Doin, M.P., Liu, Z., and Jolivet, R.: Constraining the kinematics of metropolitan Los Angeles faults with a slip-partitioning model, Geophys. Res. Lett., 43, 11192–11201, doi:10.1002/2016GL071061, 2016.**

**Galadini, F., Poli, M.E., and Zanferrari, A.: Seismogenic sources potentially responsible for earthquakes with M≥ 6 in the eastern Southern Alps (Thiene-Udine sector, NE Italy), Geophys. J. Int., 161, 739–762. doi:10.1111/j.1365-246x.2005.02571.x, 2005.**

**Romano, M.A., Peruzza, L., Garbin, M., Priolo, E., and Picotti, V.: Microseismic Portrait of the Montello Thrust (Southeastern Alps, Italy) from a Dense High-Quality Seismic Network, Seismol. Res. Lett., 1–16. doi:10.1785/0220180387, 2019.**

**TRANSALP Working Group: First deep seismic reflection images of the Eastern Alps reveal giant crustal wedges and transcrustal ramps. Geophys. Res. Lett. 29, 10. doi:10.1029/ 2002GL014911, 2002.**

**Vergne, J., Cattin, R., and Avouac, J.P.: On the use of dislocations to model interseismic strain and stress build-up at intracontinental thrust faults, Geophysical Journal International, 147, 155–162, 2001.**

---

## Author Comment (AC2) · 21 May 2020

**We are grateful for the constructive feedbacks provided by Romain Jolivet. He raises three main issues, which are addressed in detail below, answering as thoroughly as possible at each point raised. Furthermore, specific comments to individual cases of the manuscript are provided. In the following, we will repeat the referee's statements and our reply to it (in bold font).**

In this article, Anderlini et al propose to apply an approach that has been applied extensively to various tectonically active regions globally but, to my knowledge, not very often to the actively deforming areas in the alps. The authors first derive some velocity fields from GNSS and InSAR data and describe some available leveling measurements. They propose a decomposition of the InSAR velocity maps into vertical and horizontal velocity fields, which are then discussed. They move on to a very classic 2D elastic modeling of the deformation to explore potential stress accumulation when considering the active faults in the region.
In general, the paper is well written and I do not see major issues with it. However, some points need to be discussed and my comments might require a bit of work. Figures are clear (although texts could be emphasized on the maps). I see three main issues in the paper that require being fixed before publication but, after that is done, this paper will be a very interesting contribution to the discussion on how active are these frontal thrusts surrounding the Alps. I hence recommend moderate revisions and I am looking forward to see a revised version of the article. I have set major revisions in the review system because there is no intermediate step between minor and major for this journal.

Main Comments:
1 - There is very little discussion on how the selection of the data is performed to avoid the effect of subsidence in the plain. The authors propose a strict threshold of -0.5 mm/yr of vertical motion below which any deformation is considered as subsidence and removed from the data fed into the model. In my opinion, this is risky, as some long wavelength subsidence might affect the general pattern of deformation. If subsidence is high near the coast and in the plain, as implied by the data, then there should be a bending effect that will affect the whole dataset. The wavelength of such bending might depend on the processes at stake, but it is unlikely that a strict threshold will allow to bypass this discussion.

**We understand the raised point. We are aware that in this region several different deformation sources are involved besides tectonic loading. Concerning the widespread subsidence in the Venetian plain, there is an extensive literature discussing all the processes involved but no bending effects are mentioned. To justify the chosen threshold, we add in Section 3.4 an in-depth description of the ongoing processes as follows: "Several studies investigated subsidence processes in the Venetian plain (e.g. Carminati and Di Donato, 1999; Carbognin et al., 2004; Teatini et al., 2005; Bock et al., 2012), which is due to three main causes (both of natural and anthropogenic non-tectonic origin): 1- aquifer compaction after the strong groundwater withdrawal in the second half of the last century (e.g. Gatto and Carbognin, 1981; Carbognin et al., 1995); 2- uncontrolled expansion of coastal settlements and industrial activities (e.g. Tosi et al., 2002); 3- recent sediment compaction (e.g. Brambati et al., 2003; Fontana et al., 2008). As we can see from the profile A-B of Fig.5, subsidence rates increase from the center of the plain towards the coasts as due to the sum of the aforementioned processes." Since none of the these processes can**

be taken into account in our model, we choose the threshold of -0.5 mm/year and we add in the manuscript a reference to Section 3.4 to justify the choice.

My point mainly arises from the fact that (and this is an issue) your model does not really fit the InSAR and leveling data you are using. The relatively high rates of uplift measured in the north are not correctly predicted by your model (which underdetermines uplift) while the low rates to the south are over-determined. It seems that there is a constant trend between the geodetic data and the model. Geodetic data agree well with each other, which is great, but the model does not really manage to catch up. This could also be caused by isostatic adjustment adding a long wavelength deformation (i.e. a wavelength longer than profile you have established). One possibility would be to explore the effect of a linear trend (or whatever long-wavelength pattern you can think of) that would represent the long wavelength deformation needed on top of what results from dislocations in an elastic half space. This requires exploring the tradeoff between this long wavelength deformation signal and what is predicted in terms of locking depth and slip rates for both faults. It should have an impact and should be accounted for in the inverse problem.

**As concern the uplift of the Alps, the processes responsible for this signal are: 1) active tectonic shortening, which is expected to be significant in the Eastern Southern Alps due to the active compressional Adria-Eurasia convergence 2) glacial isostatic adjustment (GIA), which is expected to be higher in the center of the Alpine chain 3) erosional unloading 4) geodynamic processes due to mantle flows. We refer to Sternai et al. (2019) for a review of the aforementioned processes. Considering that for the eastern Alps the contribution to the uplift rates due to mantle flows (process n. 4) seems to be uncertain and negligible (Sternai et al., 2019), plausible estimates of the isostatic adjustment to deglaciation (n. 2) and erosion (n. 3) may account for up to ~80% of the budget of observed uplift rates in the Eastern Alps (see Fig.8 of Sternai et al., 2019).**

**Several models have been proposed to quantify the Alpine uplift due to the glacio-isostatic contribution (e.g. Barletta et al., 2006; Spada et al. 2009; Norton and Hampel, 2010; Mey et al., 2016) and the erosional unloading (e.g. Sternai et al., 2012, 2019; Mey et al., 2016), mostly by means of large scale models with a poor spatial resolution. However, it is known that these models are less reliable at the border of the Alpine ice cap (Sternai et al., 2019), where our study area is located. In the original manuscript we widely discussed about this issue in Section 6 (L380-391), and given these uncertainties, we have not considered to correct the geodetic observations for a long-wavelength isostatic contribute. However we acknowledged its importance, stating that "a possible correction for these contributions would slightly reduce the intensity of uplift rates. If it were possible to apply such a correction, the slip-rates estimates on the fault planes could be slightly reduced, in turn decreasing a little the seismogenic potential associated with the MT and BVT faults." (L389-391).**

**In order to provide a quantitative estimate, we made a test by inverting for slip-rates and locking-depths assuming the same fault geometry, but geodetic rates corrected for a long-wavelength vertical signal, which is here assumed as a linear gradient of uplift rate along 100 km of distance. Considering the mean uplift rate of 1 mm/year in the northernmost sector of the study area (the Dolomites), we choose 0.8 mm/year (80%) as maximum vertical isostatic adjustment to be removed from the observed vertical velocities. In particular we consider a linear trend (with a slope of 0.008 mm/(year*km)) that starts from Treviso city (TREV and TVSO GPS stations), along the same direction of the profile indicated in Fig. 2, and reaching the limit of 0.8 mm/year of uplift rate in correspondence of POZZ station.**

We are aware that this model is purely speculative and based on strong assumptions, but we can consider this simple approach as an upper-bound case on how considering or not these long-wavelength signal may affect the results of the inversion.

We remove this linear signal from all the geodetic datasets, i.e. from the vertical component of GPS velocities, leveling data and subsampled InSAR LoS rates, and perform the inversion of the modified velocities in order to estimate locking depths and slip rates for the proposed fault geometry. Due to these changes in the input observations, we re-evaluate the relative weighting factor *Wsar*, finding an optimal value of 0.68.

The results of the inversion are presented in the Figure A1 (below), showing the same information reported in Figure 7 of the manuscript with a few differences: the purple line in the section of the vertical rates represents the linear gradient we removed, and light gray dots indicate the unmodified original datasets, while all the other data are corrected for the linear gradient. In the bottom panel the estimated parameters (locking depth and dip-slip rates) are reported, as well as in the following table.

| Dataset | LD Montello Ramp | LD Bassano Ramp | Slip rate Montello Ramp | Slip rate Montello Flat | Slip rate Bassano Ramp | Slip rate Deep Ramp | RMSE GPS | RMSE LEV | RMSE InSAR |
|---|---|---|---|---|---|---|---|---|---|
| Corrected data | 4.6 km | 8.6 km | 0.5 mm/a | 0.35 mm/a | 2.0 mm/a | 2.3 mm/a | 0.46 mm/a | 0.57 mm/a | 0.59 mm/a |
| Original Data | $5.6_{-3.8}^{+3.5}$ km | $9.1_{-0.6}^{+1.3}$ km | $0.5_{-0.1}^{+0.2}$ mm/a | $0.4 \pm 0.1$ mm/a | $2.1_{-0.6}^{+0.8}$ mm/a | $2.5_{-0.7}^{+0.8}$ mm/a | 0.44 mm/a | 0.72 mm/a | 0.66 mm/a |

We observe that there are no substantial differences with respect to the optimal fault parameters obtained with the original dataset (Table 2 of manuscript, also reported above). This correction lead to a slight decrease of slip rates and locking depths, which are, however, all largely within the error bounds of the optimal model. We can note that at the expense of a slight increase of GPS RMSE, the misfit for the other data decreases, allowing for a better balancing among the three dataset. The slight increase of GPS residuals is mainly due to the misfit between the model and the vertical velocities to the north (see PASS and FDOS in the figure below) that depends, however, on the vertical gradient we remove which is steeper than the gradients expected from large scale models. In light of these results we do not aim at correcting the geodetic velocities for an isostatic uplift signal, considering also that the estimated fault parameters doesn't provide significant differences in terms of slip-rates and locking depths. Most of these considerations have been added in Section 6 of the main text and the specific details in the Supplement.

[Figure]

**Figure A1:** Cross sections, across the A-B profile of Fig. 2 of the manuscript, showing the modeled (black lines) horizontal and vertical velocities, as well as the SAR ascending and descending LoS rates, along with the measured ones. Green points indicate leveling data and small blue dots represent the subsampled InSAR LOS rates used during the inversion. The bottom panel reports the optimal fault geometry with dip-slip rates and locking depths estimates. BV: Belluno valley

2 - There is not enough details on how the InSAR data have been processed. Although the SBAS method is now quite known, quantitative information is required to assess the quality of the velocity field. It is not only because it correlates quite well with GPS that everything has been done right. For instance, correcting for tropospheric delays using a phase-topography correlation when trying to unravel a signal that correlates as well with topography is dangerous. One could easily mix deformation with tropospheric delays.

**We used the SARScape module of ENVI software, provided by Harris Geospatial Solutions (http://sarmap.ch/tutorials/sbas_tutorial_V_2_0.pdf), to perform SBAS analysis. The SBAS algorithm includes several steps (e.g Pasquali et al. 2014): creation of a connection graph (computing all differential interferograms from the input image stack according to the chosen criteria for temporal and geometric baselines), differential interferogram generation (spectral shift and adaptive filtering), phase unwrapping, orbit refinement and re-flattening, first estimation of the average displacement, atmospheric phase screen removal, and final estimation of the average displacement and mean ground velocity. In our study, we achieved a ground resolution of 90 m by using a multi-looking factor of 4 in range and 20 in the azimuth. All the Single Look Complex images (SLC) are coregistered in the master image geometry using a 90-m Digital Elevation Model (DEM) provided by the Shuttle Radar Topography Mission (SRTM). The topographic phase contribution was removed using the DEM, too. We applied the Goldstein filter (Goldstein and Werner, 1998) to smooth the differential phase and use precise DORIS orbits (provided by the European Space Agency) and the SRTM DEM to correct the computed interferograms from possible orbital ramps. We used the Delauney minimum cost flow (MCF) network (Constantini, 1998) along with the Delaunay method to unwrap the differential interferograms. The unwrapping coherence threshold at this stage was set to 0.3. We selected approximately several tens Ground Control Points (GCP) mainly at the borders of the processed frame, to perform the refinement and re-flattening step. Subsequently, the average displacement rate and residual height-correction factors were estimated by inverting a linear system through the Singular Value Decomposition method. Then, low-pass and high-pass spatial filters were used for the time-series images, to screen and remove the atmospheric phase component. In fact, the starting idea is that atmosphere is correlated in space but not in time. We considered two moving windows of 365 days and 1200 meters for the two filters (High and low pass). Finally, the solution of the inversion was geocoded through the used DEM. All of the final displacement measurements were obtained onto the satellite line of sight (LOS) direction and geocoded in the UTM 33N reference system.**

Furthermore, since the region has quite strong topographic gradients, unwrapping is probably challenging and there is not a word on that (which method is used for unwrapping? In general, which software is used to compute the interferograms?).

**As mentioned above, we adopted the Minimum Cost Flow (MCF) algorithm with the Delauney triangulation method. The latter helps the propagation of the unwrapping solution to reach coherent pixels also if they are separated by non-coherent areas. In fact, as you can see from in Figure 4 of the main text, the solution was able to overcome only the first mountainous chain propagating through the valleys. This was exactly due the presence of quite strong topographic gradients, as the reviewer has noticed. All the A-InSAR processing chain, as already mentioned above, was computed using the Sarscape software.**

Would it be possible to see a baseline plot?

Yes, the following figure shows the obtained connections graph and baselines distribution for the ASAR-Envisat datasets of both ascending and descending orbits. We choose interferometric pairs with a perpendicular baseline smaller than 450 meters and a temporal baseline lower than 600 days for both the orbits. The figure is now added in the Supplement.

[Figure]

**Fig. A2: (A) Considered pairs connection graph for the descending SAR orbit; (B) Considered baseline graph for the descending SAR orbit; (C) Considered connection graph for the ascending SAR orbit; (D) Considered baseline graph for the ascending SAR orbit.**

Also, is there connectivity issues within the network, considering potential unwrapping issues?

Looking at  Figure A2, we are confident that no remarkable unwrapping issues were found during such a step. Moreover, before following the inversion steps, we checked each interferogram discarding all the pairs showing clear unwrapping errors and keeping the ones with low atmospheric noise.

What is the RMS of the reconstruction of your time series?

We calculate the RMS considering one of the different possible formulas and exactly the following: $RMSError = \sqrt{1-r^2}SD_y$ where **SDy** is the standard deviation of each retrieved displacement time series and **r** is the correlation coefficient between the time series and the considered acquisitions. RMS is a measure of the fitting quality geocoded. It is the RMSE expressed in millimeters. The higher this value the worse the fitting and inversion quality. The RMS about all the time series retrieval is showed in the figure A3:

[Figure]

**Fig. A3: RMSE maps of the displacement time series for the ascending and descending orbits**

[Figure]

**Fig. A4: Chi-Square values for the ascending and descending orbits**

How linear is the time series?

Initially, we considered a linear model in the inversion step then, in a second unwrapping run, the non-linearity is estimated from the residuals obtained as difference from the unwrapped phase and the linear model. One of the estimated parameters at the end of the processing chain is the Chi-square value (Figure A4) relative to each time series. Such a value is a non-dimensional number representing how much the time series diverges from the linear behavior. Low values of the Chi-square indicates a quasi-linear trend of the displacement time series, high values for the contrary. In any case, the Chi-square test has

to be considered carefully. In fact, high values do not mean not reliable result for that area. In fact the chi-square does not represent an absolute but relative parameter, in the meaning that it does not have a predefined maximum value (i.e. the interferometric coherence parameter) but can vary depending on the type of the movement present in the study area. So if there are zones with a non-linear behavior, such areas will show high chi-square values and vice-versa. In our case, high values of chi-square are greater than 100, thus values within 10-15% of the maximum can be considered representative of pixels having mostly linear behavior.

Is there a time dependent signal?

**The retrieved patterns are mainly time independent, in the meaning that the most of the obtained ground deformation field shows a behavior tending towards the linear trend (red and orange areas in Figure A4)**

There is much more details provided for the processing of GPS data and the processing of InSAR being much less standardized than GPS these days (especially with the old Envisat data) suggests there is a lot to be added in the manuscript.

**We added much more info about the processing steps and the parameters setting adopted during the processing chain in the manuscript and in the Supplement.**

Finally, a lot of people have developed comparable methods for InSAR downsampling and they deserve some credit (see Lohman & Simons 2005, Jolivet et al 2012, 2015 or Sudhaus & Jonsonn 2009 for instance, but there is many other papers mentioning this).

**We agree with the reviewer, there is a lot of literature regarding downsampling methods of InSAR data and we add now a short description in Section 4: "Most of literature regarding downsampling methods of InSAR data analyzes coseismic and volcanic ground deformation. In these cases just a portion of the displacement map is characterized by high deformation gradient, thus the widely-used quadtree sampling method (e.g. Jónsson et al., 2002; Pedersen et al., 2003; Lohman and Simons, 2005; Metzger et al., 2011; Barnhart and Lohman, 2013) is appropriate. Indeed this algorithm reduce the number of data in order to represent the statistically significant portion of the signal (Jónsson et al., 2002) choosing a specific threshold value for the data variance in each iteration. This method has been applied also for interseismic studies (e.g. Jolivet et al., 2012; Maurer and Johnson, 2014; Xue et al., 2015) where, however, the signal-to-noise ratio of InSAR data is big enough to define an appropriate threshold value to avoid losing information of the deformation gradients. In our case, with low deformation gradients it is highly risky to apply a subsampling method that depends on the deformation signal itself. For this reason we apply an alternative method that uniformly reduce the density of pixels and the specific technical details are provided in the Supplement."**

3 - The description of the inversion procedure is incomplete. The algorithm used to find the minimum of the cost function should be, at least, named.

**We integrate the main text in Section 4 as follows: "The inversion method exploits a constrained, non-linear, derivative-based optimization algorithm (i.e. interior-point, see Byrd et al., 1999; Waltz et al., 2006). It allows to estimate the optimal parameter solution corresponding to a possible global minimum of the cost function representing the misfit between the model prediction and the geodetic measurements. These algorithms depend on the gradient and higher-order derivatives in order to guide them through misfit space, thus they can get trapped in a local minimum (Cervelli et al., 2001), providing the best results when the starting point is near the global minimum. However, in order to ensure that we find a global solution in the inversion, we tested several different initial guess founding always the same model estimate."**

Furthermore, I suspect there is some regularization of the inverse problem involved (maybe not), but please mention it.

**No, there is no regularization of the inverse problem, but we applied specific constraints to the parameter space to be investigated (such as locking depth within the elastic thickness and slip rates kinematically consistent among them). Indeed, we did not modify the relationship of the cost function (as it would be done in case of a regularization) that considers only the weighted misfit between observed and modeled velocities, but we took advantage of the specific options of the minimization algorithm described above, forcing the model to respect the imposed constraints.**

In addition, the data covariance is not described. How is it determined? One cannot follow the deal with weights if one cannot reconstruct the covariance matrix.

**We add in the manuscript in Section 4 the description of the data covariance: "The data covariance matrix is computed as follows:** $cov = \Sigma R \Sigma^T$ **where** $\Sigma$ **is the diagonal matrix of data uncertainty and *R* is the data correlation matrix, that is dimensionless, equal to one along the diagonal and the off-diagonal elements representing the correlation between each couple of data. Assuming the three geodetic dataset (GPS, InSAR and leveling) independent among them, the whole covariance matrix is composed by three independent blocks, one for each dataset. The correlation values are nonzero only for the three components of each GPS site, considering the measurements obtained by the GPS stations to be uncorrelated among them, and for the leveling data, following the approach of Árnadóttir et al. (1992). The InSAR data covariance matrix is instead diagonal with equal variance of 1 mm$^2$/year$^2$ for all the pixels."**

Then, there is a problem in the a posteriori covariance discussion. The authors mention the a posteriori covariance is derived for the linear terms while bootstrap is used for the non-linear terms. In my opinion, the covariance that is derived here is obtained considering the least squares criterion (without regularization? with regularization? Is it just GˆT Cdˆ{-1} G ?) but then, it only corresponds to a "slice" of the model space, that slice corresponding to the best non-linear parameters obtained. If so, the a posteriori covariance is greatly underestimated as it is only representative of a joint marginal of the full a posteriori PDF.

We estimated the a-posteriori covariance matrix considering the least squares criterion (without regularization), but we agree with the reviewer that this is a partial representation of the slip rate uncertainties. We consider now to use the bootstrap distributions both for the linear and non-linear model terms providing the errors at 95% confidence bounds. Please see the answer below and the further specific details of the adopted approach. The error bounds for the estimated parameters have been corrected in Table 2 and in Figure 7 of the manuscript. For a full description of the model parameters uncertainties we provide also the trade-off distributions between parameters pairs, replacing Figure S6 of the supplementary material with the Figure A5 shown below. We can observe from these distributions that the locking depth estimates do not show any correlation with the other parameters, while for the dip-slip rates the strict correlation among them is representative of the kinematic conservation constraint, for which the only parameter we can consider independent is the deep decollement slip rate (underlined label of Figure A5). We have modified accordingly the main text, discussing in Section 5 the results in terms of fault parameters error bounds from the frequency histograms and of possible correlation between parameters from the trade-off scatter plots.

[Figure]

**Fig A5: Model parameters distribution, obtained from the inversion of 1000 bootstrap re-samples of the original data (see Section 5). Top row: frequency histograms of the optimal fault parameters with the best optimal model (red line) and boundary values of 95 percentile confidence interval (green lines); see Table 2 for specific values. Other rows: scatter plots showing trade-off between parameter pairs.**

Finally, one can see in supplementary material figure S6 that the range of possible models for the locking on the Montello Ramp is bi-modal.

**Figure S6 (now, top row of Figure A5) doesn't show the range of possible models, but the collection of the optimal models we obtain randomly resampling the data by means of a bootstrap procedure used to estimate confidence intervals of the derived parameters (Segall and Davis, 1997) without making assumptions about the underlying statistics of**

errors (Amoruso and Crescentini, 2008). This method reflects the limitation of the data set used (Cervelli et al., 2001), and the bi-modal behavior of the locking depth of the Montello Ramp should be interpreted as representative of the low capability of the data to constrain this parameter, for which indeed the deformation signal is close to the techniques limits and velocity measurements appears noisy. These considerations are now added to the manuscript in Section 5 to provide a more complete description of the error bounds definition.

Then, if it is not Gaussian, why choosing the mean model?

**We didn't choose the mean model but the optimal model estimated by the inversion algorithm, described above, using the whole geodetic dataset.**

It seems that some models could be more appropriate. Would it be possible to sample for all the possible models using a Monte Carlo approach, which would give all the tradeoffs between the various parameters (and potentially solve the issue raised in my first comment)?

**The bootstrap resampling provides optimal model distribution and the tradeoffs between the various parameters as shown in Figure A5.**

For the minor comments, please refer to the annotated pdf I have sent along with my review.

**We corrected the minor comments annotated in the pdf.**

Looking forward to read an improved manuscript, if I am required to do so. I also strongly encourage the authors to add their geodetic data (i.e. the GPS, InSAR and leveling rates presented in the paper) to an online repository so other scientists can have a go at the modeling, once this study is published.

**Since the GPS and leveling rates are already available in the Supplement, only InSAR velocities are made available in an online repository.**

Please also note the supplement to this comment:
https://www.solid-earth-discuss.net/se-2020-10/se-2020-10-RC2-supplement.pdf

**References**

Amoruso, A., Crescentini, L., 2008. Inversion of synthetic geodetic data for the 1997 Colfiorito events: clues on the effects of layering, assessment of model parameter PDFs, and model selection criteria. Ann. Geophys., 51, doi:10.4401/ag-3027.

Árnadóttir, T., Segall, P., and Matthews, M., 1992. Resolving the discrepancy between geodetic and seismic fault models for the 1989 Loma Prieta, California, earthquake, Bulletin of the Seismological Society of America, 82, 2248–2255.

Barletta, V. R., C. Ferrari, G. Diolaiuti, T. Carnielli, R. Sabadini, C. Smiraglia, 2006. Glacier shrinkage and modeled uplift of the Alps, Geophys. Res. Lett., 33, L14307, doi:10.1029/2006GL026490

Barnhart, W. D., R. B. Lohman, 2013. Vertical partitioning of strain during earthquake sequences in Iran: Phantom earthquakes and triggered aseismic creep, Geophys. Res. Lett., 40, 819–823, doi:10.1002/grl.50201.

Bock, Y., S. Wdowinski, A. Ferretti, F. Novali, A. Fumagalli, 2012. Recent subsidence of the Venice Lagoon from continuous GPS and interferometric synthetic aperture radar, Geochem. Geophys. Geosyst., 13, Q03023, doi:10.1029/2011GC003976.

Brambati, A., L. Carbognin, T. Quaia, P. Teatini, L. Tosi, 2003. The Lagoon of Venice: Geological Setting, Evolution and Land Subsidence, Episodes, 26, 264-268.

Byrd, R.H., Hribar, M. E., Nocedal, J. 1999, "An Interior Point Algorithm for Large-Scale Nonlinear Programming," SIAM Journal on Optimization, Vol 9, No. 4, pp. 877–900.

Carbognin L., L. Tosi, P. Teatini, 1995. Analysis of actual land subsidence in Venice and its hinterland (Italy). In: Barends J.F., et al. (1995, eds.) - Land Subsidence, 129-137, A. A. Balkema, Rotterdam

Carbognin, L., P. Teatini, L. Tosi, 2004. Eustacy and land subsidence in the Venice Lagoon at the beginning of the new millennium. J. Marine System, 51(1–4), 345– 353.

Carminati, E., G. Di Donato, 1999. Separating natural and anthropogenic vertical movements in fast subsiding areas: The Po plain (N. Italy) case, Geophys. Res. Lett., 26, 2291–2294, doi:10.1029/1999GL900518.

Costantini, M. 1998. A novel phase unwrapping method based on network programming. IEEE Trans. Geosci. Remote Sens., 36, 3, 813-821, DOI: 10.1109/36.673674

Fontana, A., Mozzi, P., Bondesan, A., 2008. Alluvial megafans in the Veneto-Friuli Plain: Evidence of aggrading and erosive phases during Late Pleistocene and Holocene. Quaternary International, 189(1), 71–90

Gatto, P., Carbognin, L., 1981. The Lagoon of Venice: Natural environmental trend and man-induced modification. Hydrological Science Bulletin, 26(4), 379– 391

Goldstein, R., Werner, C., 1998. Radar interferogram filtering for geophysical applications. Geophysical Research Letter, 25, 21, 4035-4038, DOI: 10.1029/1998GL900033.

Jolivet, R., C. Lasserre, M.-P. Doin, S. Guillaso, G. Peltzer, R. Dailu, J. Sun, Z.-K. Shen, and X. Xu, 2012. Shallow creep on the Haiyuan Fault (Gansu, China) revealed by SAR Interferometry, J. Geophys. Res., 117, B06401, doi:10.1029/2011JB008732

Jònsson, S., H. A. Zebker, P. Segall, F. Amelung, 2002. Fault slip distribution of the 1999 Mw7.1 Hector Mine, California, earthquake, estimated from satellite radar and GPS measurements, Bull. Seismol. Soc. Am., 92(4), 1377–1389, doi:10.1785/0120000922.

Lohman, R. B., M. Simons, 2005. Some thoughts on the use of InSAR data to constrain models of surface deformation: Noise structure and data downsampling, Geochem. Geophys. Geosyst., 6, doi:10.1029/2004GC000841

Maurer, J., K. Johnson, 2014. Fault coupling and potential for earthquakes on the creeping section of the central San Andreas Fault, J. Geophys. Res. Solid Earth, 119, doi:10.1002/2013JB010741.

Metzger, S., Jónsson, S., Geirsson, H., 2011. Locking depth and slip-rate of the Húsavík Flatey fault, North Iceland, derived from continuous GPS data 2006–2010. Geophysical Journal International, 187: 564-576. doi:10.1111/j.1365-246X.2011.05176.x

Mey, J., Scherler, D., Wickert, A.D., Egholm, D.L., Tesauro, M., Schildgen, T., Strecker, M.R., 2016. Glacial isostatic uplift of the European Alps. Nature Commun. https://doi.org/10.1038/ncomms13382.

Norton, K.P., Hampel, A., 2010. Postglacial rebound promotes glacial re-advances – a case study from the European Alps. Terra Nova 22 (4), 297–302.

Pasquali, P., Cantone, A., Riccardi, P., Defilippi, M., Ogushi, F., Gagliano, S., Tamura, M., 2014. Mapping of ground deformations with interferometric stacking techniques, Land Application of Radar Remote Sensing, Holecz, F., Pasquali, P., Milisavljevic, N., Closson, D., IntechOpen, DOI: 10.5772/58225. Available online

Pedersen, R., Jònsson, S., Àrnadòttir, T., Sigmundsson, F. Feigl, K.L., 2003. Fault slip distribution of two June 2000 Mw6.5 earthquakes in South Iceland estimated from joint inversion of InSAR and GPS measurements, Earth. planet. Sci. Lett., 213(3–4), 487–502. doi:10.1016/S0012-821X(03)00302-9.

Segall, P., Davis, J. L., 1997. GPS applications for geodynamics and earthquake studies, Annu. Rev. Earth Planet. Sci., 23, 201–336.

Spada, G., Stocchi, P., Colleoni, F., 2009. Glacio–isostatic adjustment in the po plain and in the northern adriatic region. Pure Appl. Geophys. 1303–1318. https://doi.org/10.1007/s00024-004-0498-9.

Sternai, P., Herman, F., Champagnac, J.-D., Fox, M., Salcher, B., Willett, S.D., 2012. Preglacial topography of the European Alps. Geology 40 (12), 1067–1070.

Sternai, P., Sue, C., Husson, L., Serpelloni, E., Becker, T.W., Willett, S.D., Faccenna, C., Di Giulio, A., Spada, G., Jolivet, L., Valla, P., 2019. Present-day uplift of the European Alps: evaluating mechanisms and models of their relative contributions. Earth-Sci. Rev., 190, 589-604. https://doi.org/10.1016/j.earscirev.2019.01.005

Tosi, L., Carbognin, L., Teatini, P., Strozzi, T., Wegmüller, U., 2002. Evidence of the present relative land stability of Venice, Italy, from land, sea, and space observations, Geophysical Research Letters, 29(12), 1562, doi:10.1029/2001GL013211.

Waltz, R. A. , J. L. Morales, J. Nocedal, and D. Orban, 2006. "An interior algorithm for nonlinear optimization that combines line search and trust region steps," Mathematical Programming, Vol 107, No. 3, pp. 391–408.

Xue, L., S. Schwartz, Z. Liu, L. Feng, 2015. Interseismic megathrust coupling beneath the Nicoya Peninsula, Costa Rica, from the joint inversion of InSAR and GPS data, J. Geophys. Res. Solid Earth, 120, 3707–3722, doi:10.1002/2014JB011844.

---

## Author Comment (AC3) · 21 May 2020

Response to: SHORT COMMENT - GIOVANNI MONEGATO

**We are grateful for the useful comments provided by Giovanni Monegato. He raises some specific comments to which we answered point by point. In the following, we will repeat the Monegato's statements and our reply to it (in bold font).**

Dear Authors of the manuscript "New insights into active tectonics and seismogenic potential of the Italian Southern Alps from vertical geodetic velocities". This article is interesting in providing new geodetic data about deformations in this important sector of the Southern Alps. My remarks are related to the general approach that, here and in many similar works, the chronology and Quaternary stratigraphy are used for deformation rates assessment without using updated information about the landscape and sedimentary evolution of the study area. This happens because no geomorphologists or Quaternary stratigraphers are normally involved in such kind of studies. However, because of the importance of the age assessment is a key for the conclusions of these works, a more careful attention has to be paid. In the present work, many chronological references sound obsolete for the innovative scope of the research.

I listed the specific concerns for this work as follow:

Line 48: the "last deglaciation" is meant after the Last Glacial Maximum I would include the time span (19-11 ka BP according to Clark et al., 2012), for the Alps see also Ivy-Ochs (2015).

**We agree with this comment and the text has been changed accordingly, the timing for global estimation of the age of the last deglaciation is set in the time window 19-11 Ka BP following Clark a et al., 2012 and references therein.**

Line 52: Spada et al. (2009) is referred to the "last deglaciation" but not to the "present deglaciation" that refers to the ice waning after the Little Ice Age.

**Thanks, we removed the reference from the text.**

Lines 345-350: the use of old and updated terminology (Mindel-Riss-Würm) from old literature shows how the authors have not updated their knowledge about the Pleistocene stratigraphy of the study area. This was update in a review article by Carton et al. (2009), who provided a long list of radiocarbon data for the Piave catchment.

**Even if the terminology is a bit old, a possible update would not provide further information on the considerations derived in this part of the manuscript.**

Lines 353-354: the LGM is not at 20-18 ka but at 26-21 ka. I suggest to use Clark et al. (2009) for a global LGM or Monegato et al. (2017) for the Alpine LGM or Rossato et al. (2018) that gave the most updated chronology for the Veneto Prealps and piedmont plain. Also Pellegrini et al. (2005) included a useful chronology for the Belluno and Cesen-Col Visentin sector.

**We agree with this comment and the text has been changed accordingly, the timing for LGM at global scale in terms of global ice-sheet and mountain-glacier extent is now set at 26-19 Ka BP following Clark et al., 2009, while the regional alpine LGM is set at 25-23**

**Ka BP with a second large advance of the ice at 23-21 Ka BP, following Monegato et al. (2017) and reference therein.**

Line 366: maybe the Geological and geomorphological data appears thus not fully consistent just because the Authors need to use the proper updated dataset.

**We have used all the available datasets of our knowledge.**

Lines 368-369: Benedetti et al. (2000) rates for the Montello are affected by the wrong age estimation of the Montebelluna megafan, whose age was already given as pre-LGM by Fontana et al. (2008, 2014). I suggest to use these references for this discussion.

**We agree with this comment and the text has been modified taking into account that the Montebelluna megafan can tentatively be considered older than 30,000 BP, thus of pre-LGM age, but probably still within the Upper Pleistocene (125 Ka BP), following Mozzi, 2005 and Fontana et al., 2014.**

Lines 394-395: Benedetti et al. (2000) cannot be used for assessing the thickness of the Piave glacier in Belluno, because the work is not about Pleistocene glaciations. Pellegrini et al. (2005) and Carton et al. (2009) have to be considered instead. The Piave glacier was not an ice-sheet, which has to be used for the Alps (i.e., Alpine Ice Sheet).

**We agree with this comment and we modified the text accordingly taking into account that during the LGM the Belluno valley hosted the Piave glacier about 800 m thick up to an elevation of about 1150 m**

Line 401: this is age is unreliable, the deglaciation in the Piave catchment occur after 18-17 ka, see Pellegrini et al. (2005) and Carton et al. (2009). At 10 ka valley glaciers were already waned. The collapse of the Alpine glaciers was fast but it occurred at 18-17 ka as reported also by Ravazzi et al. (2014) and Rossato and Mozzi (2016).

**We understand the point, but we set up this simple model just to estimate a first order of possible uplift rate due to the melting of this glacier. Since the more recent time estimates are older than the age we choose, at most, the true uplift rates would be lower than those estimated by the model, that already predicts modest glacial isostatic uplift rates. In light of your comment we add this sentence in the manuscript: "This is an upper bound estimate since recent studies (Pellegrini et al., 2005; Carton et al., 2009) evaluate the complete disappearance of the glacier occurred likely before 15 ka BP."**

References, **in bold those used and to be added in our list**

Benedetti, L., Tapponnier, P., King, G. C., Meyer, B., & Manighetti, I. (2000). Growth folding and active thrusting in the Montello region, Veneto, northern Italy. Journal of Geophysical Research, 105(B1), 739–766.

**Carton, A., Bondesan, A., Fontana, A., Meneghel, M., Miola, A., Mozzi, P., Primon, S. & Surian, N. (2009). Geomorphological evolution and sediment transfer in the Piave River**

watershed (north-eastern Italy) since the LGM. Géomorphologié: Relief. Process. Environ. 3, 37–58.

Clark, P. U., Dyke, A. S., Shakun, J. D., Carlson, A. E., Clark, J., Wohlfarth, B., Mitrovica, J. X., Hostetler, S. W. & McCabe, A. M. (2009). The Last Glacial Maximum. Science 325, 710–714.

Clark, P.U., Shakun, J.D., Baker, P.A., Bartlein, P.J., Brewer, S., Brook, E., Carlson, A.E., Cheng, H., Kaufman, D.S., Liu, Z., Marchitto, T.M., Mix, A.C., Morrill, C., Otto- Bliesner, B.L., Pahnke, K., Russell, J.M., Whitlock, C., Adkins, J.F., Blois, J.L., Clark, J., Colman, S.M., Curry, W.B., Flower, B.P., He, F., Johnson, T.C., Lynch-Stieglitz, J., Markgraf, V., McManus, J., Mitrovica, J.X., Moreno, P.I. & Williams, J.W. (2012). Global climate evolution during the last deglaciation. Proc. Natl. Acad. Sci. 109 (19), E1134–E1142.

Fontana, A., Mozzi, P., & Bondesan, A. (2008). Alluvial megafans in the Veneto-Friuli Plain: Evidence of aggrading and erosive phases during Late Pleistocene and Holocene. Quaternary International, 189(1), 71–90.

Fontana, A., Mozzi, P., & Marchetti, M. (2014). Alluvial fans and megafans along the southern side of the Alps. Sedimentary Geology, 301, 150–171.

Ivy-Ochs, S. (2015). Glacier variations in the European Alps at the end of the last glaciation. Cuadernos de Investigación Geográfica 41, 295– 315.

Monegato, G., Scardia, G., Hajdas, I., Rizzini, F. & Piccin, A. (2017). The Alpine LGM in the boreal ice-sheets game. Sci. Rep. 7, 2078.

Mozzi, P., 2005. Alluvial plain formation during the Late Quaternary between the southern Alpine margin and the Lagoon of Venice (northern Italy). Geografia Fisica e Dinamica Quaternaria (Suppl. 7), 219–230.

Pellegrini, G.B., Albanese, D., Bertoldi, R. & Surian, N. (2005). La deglaciazione alpina nel Vallone Bellunese, Alpi meridionali orientali. Geografia Fisica e Dinamica Quaternaria. Supplemento 7, 271–280.

Ravazzi, C., Pini, R., Badino, F., De Amicis, M., Londeix, L. & Reimer, P. (2014). The latest LGM culmination of the Garda Glacier (Italian Alps) and the onset of glacial termination. Age of glacial collapse and vegetation chronosequence. Quat. Sci. Rev. 105, 26–47.

Rossato, S. & Mozzi, P. (2016). Inferring LGM sedimentary and climatic changes in the southern Eastern Alps foreland through the analysis of a 14C ages database (Brenta megafan, Italy). Quaternary Science Reviews, 148, 115-127.

Rossato, S., Carraro, A., Monegato, G., Mozzi, P., & Tateo, F. (2018). Glacial dynamics in pre-Alpine narrow valleys during the Last Glacial Maximum inferred by lowland fluvial records (northeast Italy). Earth Surface Dynamics, 6, 809-828.

Best Wishes Giovanni Monegato